# CAM-GAN: Continual Adaptation Modules for Generative Adversarial Networks

**Sakshi Varshney**[1*], **Vinay Kumar Verma**[2*], **P. K. Srijith**[1], **Lawrance Carin**[4], **Piyush Rai**[3†]

IIT Hyderabad[1], Duke Univeristy[2], IIT Kanpur[3], KAUST Saudi Arabia[4]

cs16resch01002@iith.ac.in, vv65@duke.edu, srijith@cse.iith.ac.in
larry.carin@kaust.edu.sa, piyush@cse.iitk.ac.in

## Abstract

We present a continual learning approach for generative adversarial networks (GANs), by designing and leveraging parameter-efficient feature map transformations. Our approach is based on learning a set of global and task-specific parameters. The global parameters are fixed across tasks whereas the task-specific parameters act as local adapters for each task, and help in efficiently obtaining task-specific feature maps. Moreover, we propose an element-wise addition of residual bias in the transformed feature space, which further helps stabilize GAN training in such settings. Our approach also leverages task similarities based on the Fisher information matrix. Leveraging this knowledge from previous tasks significantly improves the model performance. In addition, the similarity measure also helps reduce the parameter growth in continual adaptation and helps to learn a compact model. In contrast to the recent approaches for continually-learned GANs, the proposed approach provides a memory-efficient way to perform effective continual data generation. Through extensive experiments on challenging and diverse datasets, we show that the feature-map-transformation approach outperforms state-of-the-art methods for continually-learned GANs, with substantially fewer parameters. The proposed method generates high-quality samples that can also improve the generative-replay-based continual learning for discriminative tasks.

## 1  Introduction

Lifelong learning is an innate human characteristic; we continuously acquire knowledge and upgrade our skills. We also accumulate our previous experiences to learn a new task efficiently. However, learning new tasks rarely affects our performance on already-learned tasks. For example, after learning one programming language, if we learn a new such language it is rare that our skill on the first deteriorates. In fact, the knowledge of the previous task(s) often speeds up learning of the subsequent task(s). On the other hand, attaining lifelong learning in deep neural networks is still a challenge. Naive implementation of continual learning with deep learning models suffers from catastrophic forgetting [1, 2, 3], which makes lifelong or continual learning (CL) difficult. Catastrophic forgetting refers to the situation when a model exhibits a decline in its performance on previously learned tasks, after it learns a new task.

Several prior works [1, 3, 2, 4, 5, 6, 7, 8] have been proposed to address catastrophic forgetting in deep neural networks, with most of them focusing on classification problems. In contrast, continual learning for unsupervised deep generative models, such as generative adversarial networks (GAN) [9] and variational auto-encoders (VAE) [10] is relatively less studied so far. Some recent work [6, 11, 12, 13] has tried to address catastrophic forgetting in GANs. Among these, the generative-replay-based approach [6, 11] usually works well only when the number of tasks is small. As the number of tasks

---

*Equal contribution. † Currently with Google Research

35th Conference on Neural Information Processing Systems (NeurIPS 2021)

become large, the model starts generating unrealistic samples that are not suitable for generative replay. A regularization-based approach for continual learning in GANs [12] often gives sub-optimal solutions. In particular, instead of learning the optimal weights for each task, it learns a mean weight for all tasks, since a single set of weights tries to generalize across all tasks. Therefore, these approaches mix the sample generation of various tasks and generate blurry and unrealistic samples. Also, continually learning new tasks on a fixed size network is not feasible since, after learning a few tasks, the model capacity exhausts and the model becomes unable to learn future tasks.

Recently, expansion-based CL models [14, 15, 16, 17, 7, 18, 19, 20, 21, 22] have shown promising results for discriminative (supervised) continual learning settings. These approaches are dynamic in nature and allow the number of network parameters to grow to accommodate new tasks. Moreover, these approaches can also be regularized appropriately by partitioning the previous and new task parameters. Therefore, unlike regularization-based approaches, the model is free to adapt to novel tasks. However, despite their promising performance, the excessive growth in the number of parameters [15, 19, 20, 21, 22] and floating-point (FLOP) requirements are critical concerns.

In this work, we propose a simple expansion-based approach for GANs, which continually adapts to novel tasks without forgetting the previously-learned knowledge. The proposed approach considers a base model with *global parameters* and corresponding *global feature* maps. While learning each novel task, the base model is expanded to consider a task-specific feature transformation which efficiently adapts a global feature map to a task-specific feature map in each layer. Even though the total number of parameters increases due to the additional *local/task-specific parameters*, this feature-map transformation approach allows the leveraging of efficient architecture design choices (e.g., groupwise and pointwise convolution) to obtain compact-sized *task-specific* parameters, which controls the growth and keeps the proposed model compact.

We empirically observe that our feature map transformation is highly efficient, simple and effective, and shows more promising results compared to the weight space transformation based approaches [13]. In addition, our approach enables leveraging the task similarities. In particular, learning a new task that is similar to previous tasks should require less effort compared to learning a very different task; if we already know statistics and linear algebra, learning the subject machine learning is more easy compared to learning a completely different subject, e.g., computer architecture. Most existing CL approaches ignore the task-similarity information during the continual adaptation. We find that learning a novel task by initializing its task-specific parameters with the task-specific parameters of the most similar previous task significantly boosts performance. To this end, we learn a compact embedding for each task using the mean of the Fisher information matrix (FIM) per filter, and use it to measure task similarity. We observe that considering the task similarity information for parameter initialization not only boosts the model performance but can also be useful to reduce the number of *task-specific* parameters. While FIM has been used in regularization-based approaches for continual learning [2], we show how it can be used in expansion based methods like ours.

To show the efficacy of the proposed model, we conduct extensive experiments in various settings on real-world datasets. We show that the proposed approach can sequentially learn a large number of tasks without catastrophic forgetting, while incurring much smaller parameter and FLOP growth compared to the existing continual-learning GAN models. Further, we show that our approach is also applicable to the generative-replay-based discriminative continual learning (e.g., for classification problems). We empirically show that the pseudo-rehearsal provided by the proposed approach shows promising results for generative-replay-based discriminative models. Also, we conduct experiments to demonstrate the effectiveness of considering the task similarity in continual image generation, which we believe can lead to a promising direction for continual learning. Our contribution can be summarized as follows:

- We propose an efficient feature-map-based transformation approach to continually adapt to new tasks in a task sequence, without forgetting the knowledge from previous tasks.

- We propose a parallel combination of groupwise and pointwise $3 \times 3$ and $1 \times 1$ filters for efficient adaptation to novel tasks, and require significantly fewer parameters for adaptation.

- We propose an approach that leverages the Fisher information matrix to measure task similarities for expansion-based continual learning. We empirically demonstrate its superior performance on several datasets.

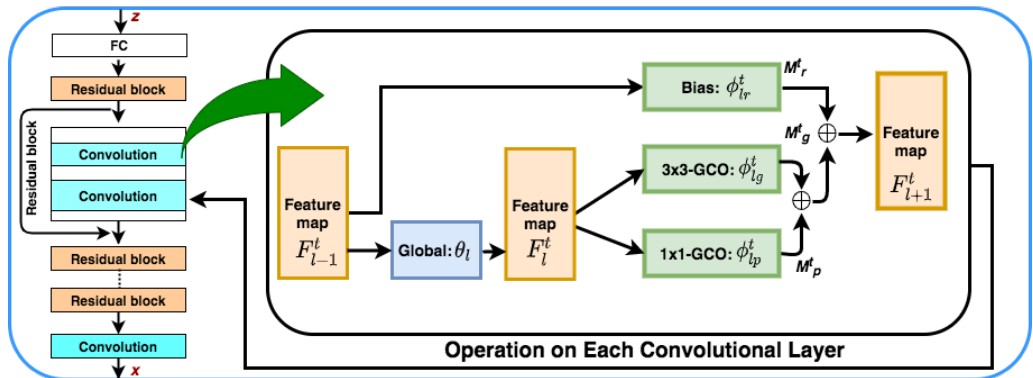

Figure 1: The proposed efficient transformation module over a convolutional layer. Each layer consists of three task-specific parameter $\phi_{lp}^t$, $\phi_{lg}^t$ and $\phi_{lr}^t$ and one global parameter $\theta_l$. Here $\phi_{lr}^t$ is the residual bias in the transformed space.

- We demonstrate the efficacy of the proposed model for continual learning of GANs and generative-replay-based discriminative tasks. We observe that, with much less parameter and FLOP growth, our approach outperforms existing state-of-the-art methods.

## 2  Lightweight Continual Adapter

**Notation and Problem Setting:**  We assume a set of tasks $\mathcal{T}_1, \mathcal{T}_2, \ldots, \mathcal{T}_K$ to be learned, and they arrive in a sequence. At a particular time $t$, data corresponding to the only current task $\mathcal{T}_t$ are available. The data for the $t^{th}$ task are given as $\mathcal{T}_t = \{x_i, y_i\}_{i=1}^{N_t}$ where $(x_i, y_i)$ represents data samples and class labels, respectively. For unconditional data generation, we assume $y_i = 1$, and for the conditional generation $y_i \in \mathcal{Y}$, where $\mathcal{Y}$ is the label set for the task $\mathcal{T}_t$.

The proposed approach consists of a base model $\mathcal{M}$ with generator parameters $\theta$, which serve as the global parameters. The global parameters at layer $l$ are denoted by $\theta_l$. Each layer of $\mathcal{M}$ also contains the task-specific parameters which perform task-specific feature-map transformation. Let $\phi_l^t$ denote the task-specific parameter for the $l^{th}$ layer during the training of task $t$. We use the GAN framework as a base generative model, based on the architecture proposed by [23], although it is applicable to other GAN architectures as well. The base model $\mathcal{M}$ comprises the discriminator (D) and generator (G). Since our main aim is to learn a good generator for each task, our continual learning approach considers only the generator to have task-specific adapters. The discriminator parameters ($\theta_d$) can be modified using each new task's data. We learn the global parameters from a base task using the GAN objective [23] and learn the task specific parameters $\phi^t$ as we adapt to a new task $t$:

$$\min_{\phi^t} \max_{\theta_d} \mathbb{E}_{\mathbf{x} \sim p_{data}^t}[log(D(\mathbf{x}; \theta_d))] + \mathbb{E}_{\mathbf{z} \sim p_{\mathbf{z}}(\mathbf{z})}[log(1 - D(G(\mathbf{z}; \theta, \phi^t); \theta_d))] + R(\theta_d, \phi^t) \quad (1)$$

where $\mathbb{E}$ denotes expectation and $R()$ is the regularization term defined in [23].

We propose an efficient expansion-based approach to mitigate catastrophic forgetting in GANs while learning multiple tasks in a sequential manner. The proposed adapter module is fairly lightweight and easy to use in any standard model for continual adaptation. The added adapter in the standard architecture may, however, destabilize GAN training. To address this issue, we add a residual adapter in the transformed feature space that results in smooth training for the modified architecture. The parameters in efficient adapters and residual bias serve as *local/task-specific* parameters and are specific to a particular task. The details of the proposed adapter and residual bias are provided in Sections 2.1 and 2.2, respectively.

### 2.1  Efficient Adapter

We propose task-specific adapter modules to continually adapt the model to accommodate novel tasks, without suffering from catastrophic forgetting for the previous tasks. The proposed approach

leverages a layerwise transformation of the feature maps of the base model, using the adapter modules to learn each novel task. While training the task sequence, the global parameters corresponding to base model $\mathcal{M}$ remain fixed. Let $F_l^t \in \mathbb{R}^{m \times n \times c}$ be the feature map obtained at the $l^{th}$ layer upon applying the global parameters $\theta_l$ of the $l^{th}$ layer during training of task $t$, i.e.,

$$F_l^t = f_{\theta_l}(F_{l-1}^t) \tag{2}$$

Here $f_{\theta_l}$ represents convolution and $F_{l-1}^t$ is the previous layer's feature map. Our objective is to transform the feature map $F_l^t$ that is obtained at the $l^{th}$ layer, but not yet task-specific, to a *task-specific* feature map using the task-specific adapters. Our task-specific adapters comprise two types of efficient convolutional operators. We combine both in a single operation to reduce layer latency.

**$3 \times 3$ Groupwise Convolutional Operation**  Let $f^g$ be the $3 \times 3$ groupwise convolutional operation ($3 \times 3-$GCO) [24] with a group size $k$ at a particular layer $l$, and defined by local parameters $\phi_{l_g}^t$. Therefore, we have each convolutional filter $c_i \in \mathbb{R}^{3 \times 3 \times k}$ in contrast to the standard convolutional filter $c_s \in \mathbb{R}^{3 \times 3 \times c}$. Here $k \ll c$, and therefore each filter $c_i$ is highly efficient and requires $\frac{c}{k}$ times fewer parameters and FLOPs. We apply the function $f_g$ on the feature map obtained by (2). For the $t^{th}$ task, let the new feature map obtained after applying $f^g$ be $M_g^t$, which can be written as

$$M_g^t = f_{\phi_{l_g}^t}^g(F_l^t) \tag{3}$$

Here $M_g^t \in \mathbb{R}^{m \times n \times c}$ is the feature map of the same dimension as the feature map obtained by (2). Therefore, we can easily feed the obtained feature map to the next layer without any architectural modifications. Note that the standard convolution $c_s$ accumulates the information across all $c$ channels of the feature map, and in each channel its receptive field is $3 \times 3$. If we reduce the group size to $k$, then the new convolution $c_i$ covers the same receptive field but it accumulates the information across only $k \ll c$ channels. The above process is highly efficient but reduces the model's performance, since it suffers from a non-negligible information loss by capturing fewer channels. To mitigate this potential information loss, we further leverage a specialized FLOP- and parameter-efficient convolutional filter, as described below.

**$1 \times 1$ Groupwise Convolutional Operation**  Pointwise convolution (PWC) is widely used for efficient architecture design [25, 26]. The standard PWC contains filters of size $1 \times 1 \times c$. It is 9 times more efficient as compared to the $3 \times 3$ convolution but still contains significant number of parameters. To increase the efficiency of the PWC, we further divide it into $z$ groups. Therefore, each convolutional filter $a_i$ contains the filter of size $1 \times 1 \times z$ and, since $z \ll c$, it reduces the FLOP requirements and number of parameters $\frac{c}{z} \times$ as compared to the standard PWC. Also, since $z > k$, it captures more feature maps and overcomes the drawback of the $3 \times 3-$GCO. Let $f^p$ be the $1 \times 1$ groupwise convolutional operational ($1 \times 1$-GCO) and $\phi_{l_p}^t$ be the $l^{th}$ layer local parameter given by the $1 \times 1$ groupwise convolutional operation. Assume that, after applying $f^p$, we obtain the feature map $M_p^t$, which can be written as

$$M_p^t = f_{\phi_{l_p}^t}^p(F_l^t) \tag{4}$$

Here $M_p^t \in \mathbb{R}^{m \times n \times c}$ also has the same dimension as the incoming feature map, and therefore we can easily apply this input to the next layer without any significant changes to the base architecture.

**Parallel Combination**  Note that local feature map adapters, i.e. (3) and (4), can be applied in a parallel manner and the joint feature map for task $t$ can be written as

$$M_l^t = M_g^t \oplus \beta M_p^t \tag{5}$$

Here $\beta = 1$ when we use $1 \times 1$ groupwise convolution operation, otherwise $\beta = 0$, and $\oplus$ is the element-wise addition. Therefore, using $\beta$, we can trade-off the model's parameter and FLOP growth with its performance. In this combination, $M_g^t$ is obtained by filters that capture a bigger receptive field and $M_p^t$ is obtained by filters that capture the information across longer feature-map sequences. Therefore, the combination of the two helps increase a model's performance without a significant increase in the model's parameters and FLOPs. Like most of the efficient architecture models [26, 27], we can apply the $3 \times 3$ and $1 \times 1$ convolutional operation in a sequential manner, denoted as: $M_l^t = f_{\phi_{l_p}^t}^p(f_{\phi_{l_g}^t}^g(F_l^t))$. Here $M_l^t$ requires two sequential layers in the model architecture

and execution of the second layer is followed by the first layer, which limits the parallelization of the model. Therefore, throughout the paper, we follow (5), i.e., the parallel combination. Again, note that the proposed adapter is added only to the generator network since the discriminator model's parameter can be discarded once the model for each task is trained.

## 2.2 Residual Bias

We append the proposed adapters to the base model; therefore the new architecture is defined by parameters $[\theta, \phi]$. The parameters $\theta$ are considered as global parameters and each task $t$ has its own *local/task-specific* parameter $\phi$. After adding the adapter layer (discussed in Sec. 2.1) to the base architecture, the model becomes unstable, and after training for a few epochs, the generator loss diverges and discriminator loss goes to zero. The most probable reason for this behaviour is that, after adding the efficient adapter layer, each residual block doubles the layer depth, and the effect of the residual connection for such a long sequence is not prominent. Therefore the model starts diverging. Another reason for the divergence could be that the $l^{th}$ layer expects the feature map obtained by the base task, but because of the local transformation, the obtained feature maps are very different, which cause the instability in the training. To overcome this problem, we learn another function $f^r$ using efficient convolutional operations with parameters $\phi_{lr}^t$. Here, $f^r$ is also a $3 \times 3$ groupwise convolution and, on layer $(l+1)^{th}$, it takes the feature map of layer $l-1$ as input. Therefore, the residual bias can be defined as: $M_r^t = f_{\phi_{lr}^t}^r (F_{l-1})$, where $M_r^t \in \mathbb{R}^{m \times n \times c}$ has the same dimension as $M_i^t$. Therefore, we can perform element-wise addition of this feature map to Eq. (5). We consider this as an element-wise bias, added to the output feature map. Moreover, note that it is like a residual connection from the previous layer, but instead of directly taking the output of the $(l-1)^{th}$ layer, it transforms the $(l-1)^{th}$ layer's feature map. We call this term as *residual bias* and the final feature map after the continual adapter and residual bias is defined as: $F_{l+1}^t = M_l^t \oplus M_r^t$. Therefore, at the $l^{th}$ layer for the $t^{th}$ task, the model has $\theta_l$ as the global parameter and $\phi_l^t = [\phi_{lg}^t, \phi_{lp}^t, \phi_{lr}^t]$ as the *local/task-specific* parameters. Here, $|\phi_l^t| \ll |\theta_l|$, i.e., the number of parameters in $\phi_l^t$ is much smaller than in $\theta_l$, which helps control the growth in model size as it adapts to a novel task.

## 2.3 Task-similarity based Transfer Learning (TSL)

Transfer learning plays a crucial role in achieving state-of-the-art performance in deep learning models. In continual learning approaches, each novel task's parameters are initialized with the previously learned model parameters. However, this can exhibit high variance in performance [28] depending on the order in which tasks arrive. Also, these approaches may have limited transfer learning ability because we may initialize the novel task's parameters with those of the previous tasks which may be very different. We observe that initializing the novel task's parameters with the most similar task's parameters not only boosts the model performance but also reduces variance in performance. It is easy to learn the task similarity for supervised learning settings and [29] explores the same for the meta-learning. In this work, we explore the task-similarity for more challenging settings, i.e., unsupervised learning connected to GANs.

We empirically observe that similar tasks share a similar gradient. Therefore, to measure the task correlation, our proposed approach leverages gradient information. In particular, we calculate the Fisher Information Matrix (FIM) of the generator parameters without using explicit label information. Let $\mathcal{L}_g$ be the generator loss w.r.t. parameter $\theta_g$. The diagonal approximation of the FIM for the task $\mathcal{T}_t$ can be defined as:

$$F_t = \text{Diag} \left( \frac{1}{N_t} \sum_{i=1}^{N_t} \nabla \mathcal{L}_g(x_i|\theta_g) \nabla \mathcal{L}_g(x_i|\theta_g)^T \right) \tag{6}$$

Here we consider $\theta_g$ as flattened, and consequently it is a vector. We use $F_t$ as an embedding to represent the task; note that it has the same dimension as the generator's parameters. However, directly learning and representing task embeddings in such a manner is expensive, both computationally as well as in terms of storage. We consider the convolutional filters used in the adapter modules for calculating FIM to reduce the number of parameters and we further replace each convolutional filter by its mean value, and ignore the fully connected layers to learn a compact task embedding. We use these learnt task embeddings to calculate task similarities among various tasks. Based on the similarity measure, we initialize adapter modules parameters of the current task with the parameters of the most similar previously seen task.

## 3 Related work

Catastrophic forgetting [3, 1, 30, 31, 32, 33, 8, 34, 6, 13] is a fundamental problem for deep neural networks, which arises in settings where, while learning from streams of data, the data from previous tasks are no longer available. There is growing interest in mitigating catastrophic forgetting in deep neural networks. Existing methods can be broadly classified into three categories, based on regularization, replay, and expansion. The regularization-based approach [31, 35, 30, 36, 37] regularizes the previous task's model parameters such that we learn to solve the new task with a minimal shift in the previous task's parameters. The replay-based approach [33, 8, 38, 34, 39] stores a subset of the previous tasks' samples in a small memory bank or learns a generative model via VAE [10] or GAN [9] to replay the previous tasks' samples. Dynamic expansion-based approaches [19, 14, 20, 40, 7, 41] increase the model capacity dynamically as each new task arrives, and show promising results compared to fixed-capacity models.

Most of the above-mentioned approaches focus on eliminating catastrophic forgetting in discriminative models (supervised learning, such as classification). Despite the wide popularity of deep generative models, continual learning for such models is relatively under-explored. MeRGAN [6] proposed a generative-replay-based approach to overcome catastrophic forgetting in GANs. For generative replay, MeRGAN stores the immediate copy of the generator to provide the replay of previously learned tasks, while training the new task. While generative-replay-methods succeed in solving catastrophic forgetting to some extent, it is a costly operation as it preserves a copy of the complete model parameters. It also results in blurry samples for previous tasks as the number of tasks increases. Lifelong GAN [12] performs knowledge distillation among multiple realizations of the model to prevent catastrophic forgetting in image-conditioned generation via GANs. These regularization-based models converge to the sub-optimal solution and are unable to model large task sequences. GAN-memory [13] is another recently proposed method based on expansion. It performs style modulation to learn new tasks by performing a transformation over weight parameters. GAN-memory shows promising results for continual learning; however, expansion cost and model complexity may be a bottleneck. In contrast, we propose a simple expansion-based model which is highly parameter and FLOP efficient, by employing efficient feature map transformations. It also requires minimal changes to the base model to adapt to the novel task(s). Therefore, without exploding the model size, we can learn any number of tasks in a sequence. Moreover, unlike existing continual GAN approaches, our approach can leverage task similarities (computed using an FIM based embedding for each task), bringing in an explicit capability of transfer learning [42, 43, 44, 45, 46, 47, 48, 49] within a continual learning setting.

## 4 Experiments

We perform extensive experiments on several image datasets from, visually diverse domains, to show the efficacy of the proposed approach. We perform experiments for generation of samples for continually steamed datasets. We also demonstrate that the proposed continual generative model can be used for generative-replay-based continual learning for discriminative models (supervised learning). We refer to our approach as CAM-GAN (Continual Adaptation Modules for Generative Adversarial Networks). Our code is publicly available at `https://github.com/sakshivarshney/CAM-GAN`.

### 4.1 Data Description

**Unconditional Generation:** We conduct experiments over perceptually distinct datasets to show the continual adaptivity of our model for the data-generation task. For continual data generation, we consider 7 datasets from perceptually distant domains: CelebA ($\mathcal{T}_0$) [50], Flowers ($\mathcal{T}_1$) [51], Cathedrals ($\mathcal{T}_2$) [52], Cat ($\mathcal{T}_3$) [53] Brain-MRI images ($\mathcal{T}_4$) [54], Chest X-ray ($\mathcal{T}_5$) [55] and Anime faces ($\mathcal{T}_6$)[1]. We consider $256 \times 256$ resolution images for all the datasets.

**Conditional Generation:** We also experiment on four task sequences comprising four Imagenet types of data [56]: ($i$) fish, ($ii$) bird, ($iii$) snake and ($iv$) dog. Each group contains six sub-classes; so we have 24 classes for conditional generation. In our setting, we consider each group as a task. Therefore, we have four tasks: $\mathcal{T}_1, \mathcal{T}_2, \mathcal{T}_3$ and $\mathcal{T}_4$, corresponding to each group. Each task is

---

[1] `https://github.com/jayleicn/animeGAN`

formulated as a 6-class classification problem. We selected $256 \times 256$ resolution images of selected Imagenet classes.

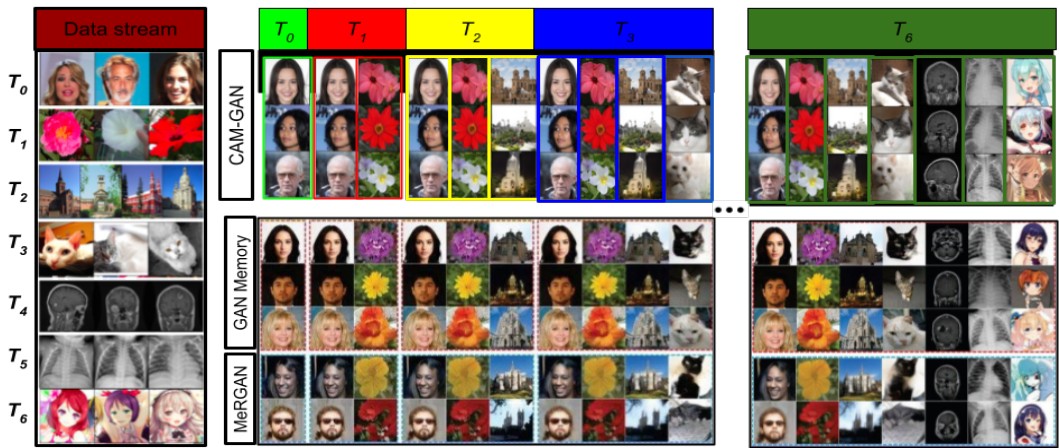

Figure 2: (Left) The real data stream representing tasks in the sequence, (right) Qualitative comparison of samples generated in streamed manner using our approach after training each task $(\mathcal{T} = \mathcal{T}_1, \mathcal{T}_2, \ldots \mathcal{T}_6)$ with GAN-memory and MeRGAN.

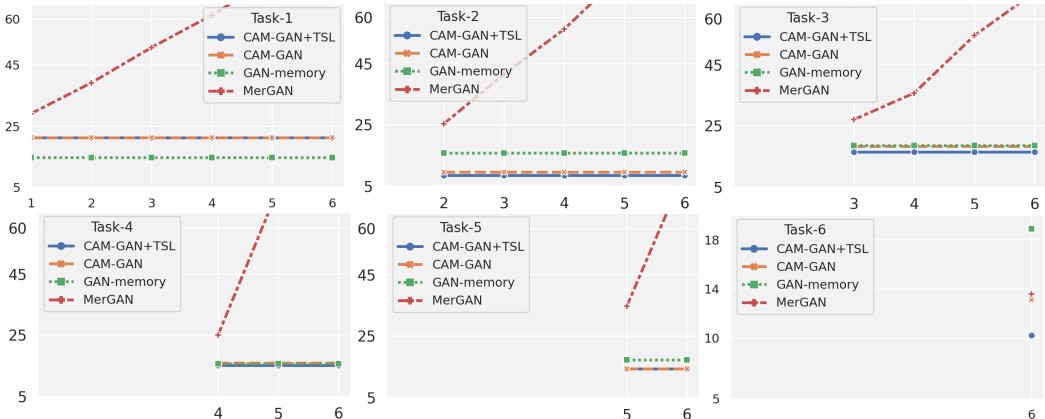

Figure 3: Comparison of FID scores of our approaches (CAM-GAN+TSL,CAM-GAN) with GAN-memory and MeRGAN. Each subgraph represents the FID of current task $\mathcal{T}$ and affects its FID after training subsequent tasks in the sequence. Our approach continually learns new tasks without catastrophic forgetting and, except for the first task, achieves lower/comparable FID scores as compared to GAN-memory and MerGAN, with substantially lower number of parameters.

## 4.2 Architectural and Evaluation Metric

We utilize the GAN architecture from GP-GAN [23]. On the top of GP-GAN, we add the proposed adapter module. We use Fréchet Inception Distance (FID) [57] as the evaluation metric to show quantitative results, as it correlates with human evaluation and is efficient to compute. Architecture and evaluation details are discussed in the Supplemental Material.

## 4.3 Unconditional Continual Data Generation

The proposed approach shows promising results for unconditional continual data generation without catastrophic forgetting. We perform the experiments over a sequence of diverse datasets to illustrate our approach's capability in the generation of samples belonging to the completely varied domains. For the unconditional generation experiments, we first train a base model $\mathcal{M}$ on the task $\mathcal{T}_0$. We consider the CelebA dataset as the $0^{th}$ task, to train the base model. The parameters of the model

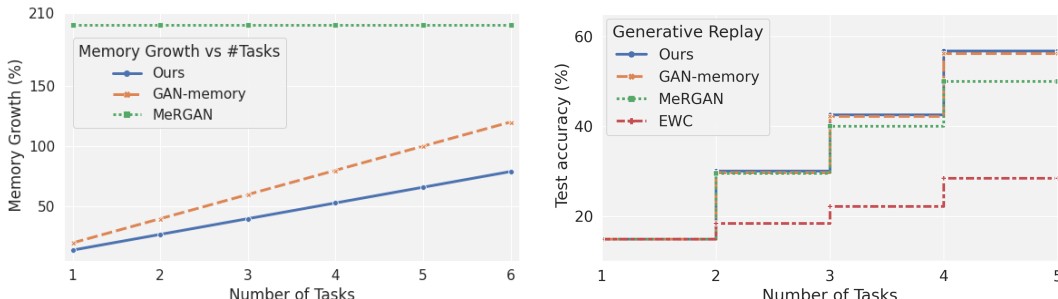

Figure 4: **Left:** Parameter Growth per task. **Right:** Comparison of combined test accuracy while using our approach as generative replay with baseline models on the supervised CL setting.

trained on $\mathcal{T}_0$ serve as global parameters. Once we have the global parameters trained on the base model, they are frozen for the subsequent task sequence $\mathcal{T}_1, \mathcal{T}_2, \ldots, \mathcal{T}_6$. Only task-specific parameters are free to change for the later tasks. We compare our results with state-of-the-art baselines, both expansion-based [13] (GAN-Memory) and generative-replay-based [6] (MeRGAN). We provide both qualitative and quantitative comparison of our model with these baselines.

Figure 2 shows the qualitative comparison of our method, CAM-GAN, with the baselines. We can observe that MeRGAN starts generating blurry samples for previous tasks with the increased length of the task sequence. In contrast, using our approach, the quality of samples for previous tasks remains un-

|  | Parameter (M) | FLOPs (G) | Parameter (% ↑) | FLOPs (% ↑) |
|---|---|---|---|---|
| Original | 52.16 | 14.75 | – | – |
| GAN-Memory | 62.69 | – | 20.18 | – |
| CAM-GAN | 58.88 | 19.69 | 12.88 | 35.55 |

Table 1: The FLOP and parameter growth for each task compared to the base model and GAN-memory [13]. Our approach has much fewer parameter growth while providing better results.

changed after training subsequent tasks in the sequence. Our simple feature map transformation requires only about $\sim 13\%$ task-specific parameters, while GAN-Memory requires about $\sim 20\%$ parameter growth per task (Table 1; Fig. 4 (Left). In addition, as we can observe from Fig. 3 (FID score comparisons), our model generates considerably better samples for all the datasets, except the flower task.

**Incorporating Task Similarity:** We further incorporate the task-similarity measure to improve the continual adaptation of the model. Figure 6 (left) depicts that generation improves when the parameters are initialized using the task similarity metric to find the most similar previous task. Our qualitative results (Fig. 2) are supported by FID scores presented in the Fig. 3. The detailed Tables for Fig. 3 are given in the Supplemental Material. The generation of Anime dataset improves significantly when the adapter parameters are initialized using celebA parameters, instead of X-ray parameters.

### 4.4 Conditional Data Generation as Replay

Our continual GAN approach can also be used as a generative replay [38] in discriminative continual learning. We further test our model's ability in the generative replay paradigm to assist the classifier model in lifelong learning. We consider conditional data generation for generative replay experiments and apply it to the class incremental setup [58] for classification. We perform joint testing [13], i.e., while training task $t$ in the sequence, the classifier must classify all the $6 \times t$ classes accurately. We compare our method with other generative-replay approaches, MeRGAN and GAN-memory, and also compare with the regularization based approach EWC [31].

Figure 4 (right) represents the test accuracy compared to the baseline models. We observe that: $(i)$ EWC does not perform well in the incremental-class setup; its performance for the previously trained tasks declines sharply after training subsequent tasks. $(ii)$ MeRGAN provides good accuracy for the initial tasks in the sequence, but its performance starts degrading as more tasks are added in the sequence. In contrast, our approach maintains the test accuracy for previous tasks by providing good quality samples. $(iii)$ Our approach achieves better/comparable test accuracy than GAN-Memory. However, our approach uses a significantly less number of parameters to attain better/similar accuracy,

making it more scalable for generative-replay scenarios with a large number of tasks in the sequence. In the Supplemental Material, we provide model details, explain the training procedure, provide more qualitative results, and interpolation results when moving from one task to the next task.

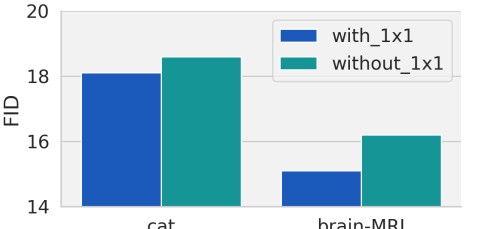 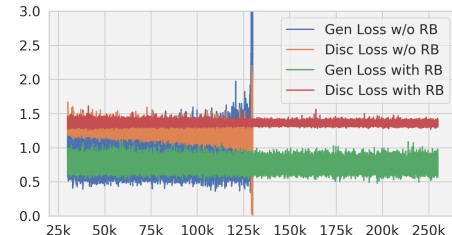

Figure 5: **Left:** Effect of $1 \times 1$ convolution on FID scores. **Right:** Effect of the Residual Bias (RB) on the Generator and discriminator loss for the CELEB dataset. The added element-wise residual bias provides highly stable training.

# 5 Ablation Study

In this section, we disentangled the various components of our model, and we observe that each proposed component plays a significant role in the model's overall performance.

## 5.1 Effect of $1 \times 1$ convolution

We perform an ablation study to demonstrate the effectiveness of $1 \times 1$ efficient convolutional operators described in Sec. 2.1 to the adapter modules. We conduct our experiments on datasets ($\mathcal{T}_3$) and brain MRI images ($\mathcal{T}_4$). For the $\mathcal{T}_3$ and $\mathcal{T}_4$ experiment, we select the previous task's model, i.e., $\mathcal{T}_2$ and $\mathcal{T}_3$, respectively, and freeze the $1 \times 1$ convolutional layer. Therefore, the $3 \times 3 - GCO$ are only free to adapt to the current task. The results are shown in Fig. 5 (left) which demonstrate that having a small portion of the total parameter ($\sim 1\%$) in the $1 \times 1$ adapter improves the performance, and the drop without this adapter is significant.

## 5.2 Effect of the Residual Bias

Section 2.2 describes the detail of the residual bias. The residual bias plays a crucial role in the model's training stability. Empirically, we observe that after inclusion of adapter modules in the GAN [23] architecture, the model shows highly irregular training, and discriminator and generator losses diverge quickly. To overcome this problem, we learned the residual bias for the feature map space. Residual bias is similar to the residual connection [59] but in the transformed feature space (since two layers are of different dimensions and element-wise addition is not feasible) with the help of learnable parameters. Figure 5 (right) shows that, without residual bias on the CelebA dataset, the discriminator and generator losses have high variance, and after $125K$ iterations, the model diverges. However, residual bias provides highly stable training as the discriminator and generator losses oscillate in a narrow range; therefore, the model is stable and does not diverge as training progresses. We train on the same dataset for $600K$ iterations for an extreme evaluation, and observe the same stability in training and loss curve without any degradation.

## 5.3 Effect of Task Similarity

If the next task tends to be very similar to the previous task, adaptation is expected to be easy and only a few *task-specific* parameters would be sufficient for adaptation and training should converge quickly. Previous approaches [6, 11, 12, 13], while learning a novel task, initialize the model from the previously learned task's parameters. The previous task may or may not be similar to the novel task, and therefore the model's performance is not optimal; in this setting one also requires a significant number of new learnable parameters and many epochs for convergence. In our approach, we measure the task similarity (discussed in Section 2.3) of the new task with all the previous tasks, and the model is initialized with the parameters of the most similar previous task. Results are shown in Fig. 6 (left) which shows that using similarity information significantly improves performance. Compared to

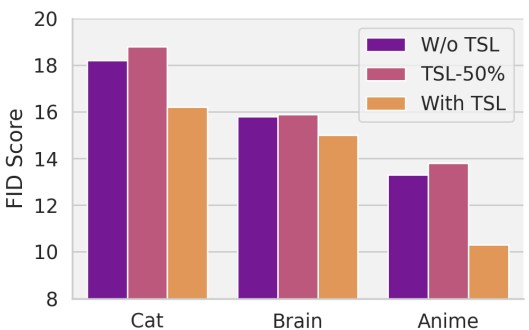 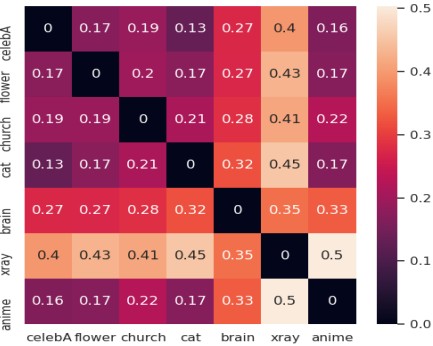

Figure 6: **Left:** Model's FID score using TSL based initialization, TSL-50% shows if we reduce the model parameters 50% and initialize it with task-similarity. **Right:** Task-similarity measure over various dataset (lower if better) using the model described in Sec-2.3

the previous task's parameter initialization, if we reduce the parameter by $\sim 50\%$, similarity-based initialization still achieves similar performance. Further details of training and hyperparameters are discussed in the Supplemental Material.

## 5.4 Effect of the Base Task

In our approach, global parameters are trained on the base task and are fixed across the subsequent tasks. The model's performance on upcoming tasks depends on the parameter robustness learned on the base task. Therefore base task plays a significant role for future learning. We perform ablation over a reasonable and weak base task to train the global parameters of our model. The ablation results are shown in Table 2. We observe that our approach can provide similar or better results with a reasonable choice

| Base task | Datasets | | |
|-----------|----------|--------|------|
| | Flower | Church | Cat |
| Lsun-Bedroom | 24.1 | 8.6 | 17.3 |
| Brain-MRI | 48.02 | 32.16 | 40.2 |

Table 2: FID score of various tasks while considering Lsun-Bedroom, Brain-MRI Datasets as base task.

of the base task. Changing the base task from CelebA to LSUN-Bedroom, our model still provides comparative or better performance on all tasks. Further, we changed the base task to a weak or very different base task, i.e. Brain-MRI dataset (grayscale medical image). Using the Brain-MRI as base task model, the performance degraded but the results are still fairly reasonable. In case of a weak base task, we can improve the model performance by increasing the number of local parameters per task.

## 6 Conclusions

We propose a simple and efficient, dynamically expandable model to overcome catastrophic forgetting in continual learning of GANs. The proposed model contains a set of *global* and *local/task-specific* parameters. The global parameters are obtained from the base task dataset and remain fixed for the remaining tasks. Each task's specific parameters act as local adapters and efficiently learn *task-specific* feature maps. The local parameters leverage parallel combinations of efficient filters that minimize the per-task model's FLOP and parameter growth. Also, to overcome the unstable GAN training, we propose a residual bias in the transformed space, which provides highly stable training. The task-similarity-based initialization significantly boosts the model's performance and provides room to further reduce the model parameters. Our approach mitigates catastrophic forgetting by design; while learning the novel task, the previous model's parameters are unchanged. Therefore, it does not show any catastrophic forgetting. We demonstrate the effectiveness of our approach through qualitative and quantitative results. The proposed model requires substantially fewer parameters ($\sim 36\%$ fewer parameters) for continual learning in GAN, while showing a better or similar result compared to the current state-of-the-art models. We also show that the proposed model can be used for generative replay, and it offers a promising way to overcome catastrophic forgetting in discriminative (supervised) models. It requires a reasonable base task to learn the global parameters. Although our experiments show that our model is fairly robust to the choice of base model, some care should be taken since a considerably weak base task may degrade the model performance.

## Acknowledgement

Sakshi Varshney acknowledges the support from DST ICPS and Visvesvaraya fellowship. The portion of this research performed at Duke University was supported under the DARPA L2M program. PR acknowledges support from Visvesvaraya Young Faculty Fellowship.

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
