# Supplementary: CAM-GAN: Continual Adaptation Modules for Generative Adversarial Networks

**Sakshi Varshney**[1*], **Vinay Kumar Verma**[2*], **P. K. Srijith**[1], **Lawrance Carin**[4], **Piyush Rai**[3†]

IIT Hyderabad[1], Duke Univeristy[2], IIT Kanpur[3], KAUST Saudi Arabia[4]

cs16resch01002@iith.ac.in, vv65@duke.edu, srijith@cse.iith.ac.in

larry.carin@kaust.edu.sa, piyush@cse.iitk.ac.in

## Appendix of CAM GAN

## 1 Preliminary

**Generative Adversarial Network (GAN):** We briefly introduce GAN [1, 2] for which our proposed a lightweight continual learning scheme. GAN is a generative model consisting of two networks; a discriminator network $\mathcal{D}$ and a generator network $\mathcal{G}$. The generator focuses on emulating the true data distribution, while the discriminator network focuses on efficiently differentiating the true data samples from the samples generated by the generator network, thus forming a zero-sum game architecture. The GAN objective function can be formulated as:

$$\mathbb{E}_{\mathbf{x} \sim p_{data}} log(D((\mathbf{x}; \theta_d))) + \mathbb{E}_{\mathbf{z} \sim p_{\mathbf{z}}(\mathbf{z})} log((1 - D(G(\mathbf{z}; \theta_g); \theta_d)))] \tag{1}$$

## 2 Experimental details

We conduct extensive experiments to show the efficacy of our approach in generation of various tasks in streamed manner. We also demonstrate that our approach can be used to perform pseudo-rehearsal of tasks to assist discriminative models to incorporate continual learning. Our approach leverages feature space for style modulation to adapt to the novel task. While performing experiments, we observe that our approach requires significant less number of iterations to adapt to new task than approaches which rely on weight space to adapt to the new task.

### 2.1 Unconditional

We train our model on various datasets to show the effectiveness of our approach in generating high-dimensional and diverse domains images in a streamed manner. We conduct experiments on Flower [3], Cathedrals [4], Cat [5], Brain-MRI images [6] , Chest X-ray images [7], and Anime faces [1]. The model $\mathcal{M}$ is trained on CelebA [8] which serves as global parameters for the remaining tasks in the sequence. We show robustness of our model to the source task, we also consider LSUN-Bedroom[4] dataset as base task.

Due to limited space, we could only demonstrate part of the generated images in the main paper(Sec.2.3). To provide better clarity about the perceptual aspects of our results, Figures 1, and 2 present the qualitative results of our approach. As evidenced from the figures, our approach generates quality samples for current as well as previously learnt tasks. Our approach leverages feature maps to preform style modulation unlike other approaches which rely on weight space transformations. Empirically, we observed that while training the model for a new task in the data stream, our approach requires significantly less number of training iterations. Please refer to Table-1 for the training iterations used in the proposed model compared to the GAN-Memory [9]. We can observe that for

---

[1] https://github.com/jayleicn/animeGAN

the flower and X-Ray dataset, our approach requires $20K$ iteration to train while GAN-Memory [9] requires $60K$ iteration. We observe a similar pattern for the Brain-MRI dataset where we require $30K$ iteration compared to GAN-memory which trained for the $60K$ iteration.

## 2.2 Continual Data generation as Replay

We further perform experiments to show efficacy of our approach as generative replay to assist the discriminative model to inculcate continual learning. We train our model for four task sequences comprising of four groups of Imagenet dataset [10] (fish,bird,snake,dog). These groups further contain six sub-classes. In generative replay framework for discriminative models, we first train the model to generate a sequence of historical tasks using our method, GAN-memory, and MeRGAN. $i$). While training the classifier network for current task $\mathcal{T}_t$, we generate samples from all the previous tasks $\mathcal{T}_1 \ldots \mathcal{T}_{t-1}$ using the continually trained model. We combine these generated samples with the training samples of the current task in the sequence to train the classifier network. We perform labelled conditional generation, as the selected dataset comprises of defined sub-classes within each task. We have used base model trained on the CelebA dataset for conditional generation setup for our approach and baseline models. We demonstrate perceptual results of conditional generation for generative replay in Figure 3. The batch size for EWC is set as $n$=24, and for replay based methods, while training task $t$, the batch size is set to $t \times n$. We select ResNet18 model as classifier network, and utilizes the ResNet-18 model pretrained on Imagenet dataset. Adam optimizer with learning rate $5 \times 1e^-5$, and coefficient $(\beta_1, \beta_2)$= (0.9,0.999) has been used for training of ResNet-18 model.

## 2.3 Architectural Details

We inherit GAN architecture from "Which Training Methods for GANs do actually Converge?"(GP-GAN)[2] [11]. We select GP-GAN architecture as it has been very successful in generating quality samples in high-dimensional spaces, by providing stable training. We use $R_1$ regularizer defined in GP-GAN paper, which regularizes the discriminator network when it deviates from the Nash equilibrium of zero-sum game defined by GAN networks. It penalizes the gradient of a discriminator for the real data term alone to provide training stability. The $R_1$ regularizer is given by

$$R_1(\psi) = \frac{\gamma}{2} E_{P_D(\mathbf{x})}[||\nabla D_\psi(x)||]^2$$

where $\psi$ represents discriminator parameters and $\gamma$ is a regularization constant. In our experiments we use $\gamma = 10$.

We first train the network on celebA $\mathcal{T}_0$ dataset with resolution 256x256. The base model's parameters trained on CelebA serve as global parameters for model $\mathcal{M}$. We have also shown robustness of our approac to the base task, We further train our model considering lsun-Bedroom dataset image generation as base task. Table 2 support our claim, our model achieves comparable FIDs for all the tasks in the sequence even considering LSUN-bedroom[4] as base task. We annex adapter modules in all convolutional layers of generator network except the last layer. We annex the adapter module in model trained on the first task also to provide better training stability across multiple tasks. The discriminator network architecture is kept similar to the architecture used in GP-GAN. We keep all the discriminator network parameters unfrozen so that the discriminator network can enforce the generator network for high-quality samples generation.

We use Fréchet Inception Distance (FID) [12] as evaluation metric to show quantitative results, as it correlates with human evaluation and is efficient to compute. FID between the two distribution $\mathcal{X}_1$ and $\mathcal{X}_2$ is defined as

$$FID(\mathcal{X}1, \mathcal{X}2) = ||\mu_1 - \mu_2||_2 + Tr(\Sigma_1 + \Sigma_2 - 2(\Sigma_1\Sigma_2)^{\frac{1}{2}})$$

where $\mathcal{X}1$ and $\mathcal{X}2$ represents a set of real images and generated images, respectively. For calculating FID, we select min($10000,|D_s|$), here $|D_s|$ represents the total images in the chosen dataset. The lower value of FIDs implies better generation, and we provide an FID score comparison with the recent approaches for continual learning in GANs.

---

[2]https://github.com/LMescheder/GAN_stability

# 3 Algorithm

---

**Algorithm 1** Task similarity

---

**Require:** Task sequence $\mathcal{T}_t$: $t = 1 \ldots K$
1:  **for** $(t = 1, \ldots, K)$ **do**
2:      Calculate task embedding $\mathbf{v}_t$ corresponding to each task $\mathcal{T}_t$.
3:      Store the task vector $\mathbf{v}_t$.
4:      **if** $t > 1$ **then**
5:          **for** $i = (t - 1) : 1$ **do**
6:              Calculate task similarity $TS(\mathbf{v}_t, \mathbf{v}_i)$
7:          **end for**
8:          Find the task $\mathcal{T}_c$ from the historical task sequence($\mathcal{T}_{(t-1)} \ldots \mathcal{T}_1$) having maximum task similarity with the current task.
9:          Initialize the adapter modules with the adapter parameters of the Task $\mathcal{T}_c$.
10:         Train the task specific parameters for the current task in the sequence.
11:     **end if**
12: **end for**

---

# 4 Task-similarity based Transfer learning (TSL)

We know that transfer learning is the primary ingredient to improve the model performance. However transfer learning with most closest task not only improves the model's performance but also requires very few parameters to learn and provides a fast convergence. Therefore it further provides a room to reduce the model's local parameters.

The detailed model are discussed in the main paper, where we calculate the FIM of the model for each task. To calculate the FIM we consider the generator loss and it does not requires the samples label. The size of the FIM embedding is same as number of parameters in the model, therefore to save the embedding for each task we requires significant storage. We empirically observe that instead of learning embedding of the whole parameters only the mean fisher information of each filter in adapter module is sufficient to capture the task information. Therefore discarding the fully connected layer and calculating the mean fisher information for each convolutional filter in adapter modules provides a close approximation for the complete FIM. The above approximation drastically reduce the embedding size and requires a negligible storage space. For each task we estimate the FIM embedding and save in the memory, when novel task arrives we trained the GAN for the few epoch and learn the FIM embedding for the generator loss. The current embedding is used to measure the task similarity with all previously learned task and initialize the model parameters with the most similar task's parameters. In the Table-5 we provide the similarity score for each task with all the previously learned task. Also, in the Table-3 we provide the results of the model with similarity based initialization. We can observe that similarity based initialization significantly boost the model performance. Also, TSL based initialization further provides a room to compress the model growth. In the Table-3, CAM-GAN (TSL) (50% params) performs only half parameters growth compared to

| Task | Flower | Cathedral | Cat | Brain-MRI | X-Ray | Anime Faces |
|---|---|---|---|---|---|---|
| GAN-Memory [9] | 60k | 60k | 60k | 60k | 60k | 60k |
| Ours | 20k | 60k | 60k | 30k | 20k | 60k |

Table 1: Number of iterations required to train novel task in the streamed sequence.

| Task | Flower | Cathedral | Cat | Brain-MRI | X-Ray | Anime |
|---|---|---|---|---|---|---|
| GAN-Memory [9] | 14.8 | 15.6 | 18.2 | 15.59 | 17.18 | 13.29 |
| CAM-GAN (CelebA) | 23.0 | 9.52 | 18.21 | 15.6 | 14.24 | 13.1 |
| CAM-GAN (LSUN) | 24.1 | 8.63 | 17.3 | 14.96 | 11.8 | 14.2 |

Table 2: Quantitative comparison (FID score) of the images generated for the task sequence using GAN-memory and CAM-GAN (using CelebA and Lsun-Bedroom as base task)

the CAM-GAN (TSL) and we can observe that it shows highly competitive result with CAM-GAN (w/o TSL) with 100 % parameters growth. Therefore using only $50\%$ parameters in the adapter with TSL based initialization we can achieve the same FID score as we obtained using the CAM-GAN with previous task initialization.

|        | CAM-GAN (w/o TSL) | CAM-GAN (TSL) | CAM-GAN (TSL) (50%params) |
|--------|-------------------|---------------|---------------------------|
| Cat    | 18.2              | 16.4          | 18.8                      |
| brain  | 15.8              | 15.0          | 15.9                      |
| Anime  | 13.3              | 10.9          | 13.8                      |

Table 3: FID score based on the similarity measure. Here TSL (50% params) represents the model with half parameter growth compared to the TSL.

|         | CAM-GAN (100% params) | CAM-GAN (50% params) |
|---------|-----------------------|----------------------|
| Flower  | 23.0                  | 35.67                |
| Church  | 9.52                  | 14.01                |
| Cat     | 18.21                 | 21.17                |

Table 4: FID score of various tasks using CAM-GAN(100% parameters), CAM-GAN(50% params). Here CAM-GAN(50% params) represents the model with half parameters to the actual architecture considered in our approach.

|           | CelebA   | Flower  | Cathedral | Cat     | Brain-MRI | X-ray   | Anime   |
|-----------|----------|---------|-----------|---------|-----------|---------|---------|
| CelebA    | 0.00000  | ———     | ———-      | ———     | ———       | ———-    | ———-    |
| Flower    | **0.17347** | 0.00000 | ———     | ———     | ———       | ———     | ———     |
| Cathedral | **0.19235** | 0.19719 | 0.00000 | ———     | ———       | ———     | ———     |
| Cat       | **0.13226** | 0.17392 | 0.21314 | 0.00000 | ———       | ———     | ———     |
| Brain-MRI | 0.27495  | **0.27205** | 0.28336 | 0.32030 | 0.00000   | ———     | ———     |
| X-ray     | 0.39600  | 0.42941 | 0.41496 | 0.45344 | 0.35121   | ———     | ———     |
| Anime     | **0.16374** | 0.17057 | 0.22379 | 0.16971 | 0.33241   | 0.50362 | 0.00000 |

Table 5: Similarity score obtained using the FIM for the task sequence.

## 4.1 Experimental details

We firstly train the model for small number of iterations to learn the FIM corresponding to current task $\mathcal{T}$, In our case, we train the model for $K = 1000$ iterations to learn the FIM for each task. While training task $t = i$, we calculate task similarity of the current task from all the previously stored task embeeding. We initialize the adapter parameters of the current task with the most similar previous task. We fine-tune the pretrained base model for finding FIM corresponding to each task. While calculating FIM may require few additional iterations, it not only boosts the model performance but also achieves the required performance in significantly fewer iterations if tasks are highly correlated. We further performed experiments by reducing the adapters parameter by half while considering the similar task for initialization. We only provide additional adapter parameters in the second convolutional layer of residual block, while considering the adapters of first convolutional layer of residual block same as the previous similar task. The results are provided in table 3. Because of the task similarity based initialization we can improve the model performance and achieved the similar model performance even further reducing the 50% parameters. In the Table-4 we have shown the results of CAM-GAN by reducing the 50% model parameters when we don't follow the task similarity based initialization. We can observe without TSL, CAM-GAN's performance significantly degraded with the reduced parameter.

## 5 Interpolation between two tasks

Our method exhibits smooth interpolations among various task generation processes. We can transfer images of one task to the other by providing weighted combination of parameters trained for

respective tasks using our approach. We require following steps to be performed for interpolating images belonging to various tasks:

- We first train our model for the datasets considered for interpolation (e.g. church, flowers).
- We take weighted combination of the task-specific parameters to perform the interpolation. Let $\phi_i$ and $\phi_j$ be the adapter parameters for the task $i$ and $j$. Then the interpolation parameters ($\phi_{interp}$) between two task can be defined as:

$$\phi_{interp} = \lambda * \phi_i + (1 - \lambda) * \phi_j \tag{2}$$

  where $\lambda \in [0, 1]$ defining the relative weight in this combination. If $\lambda = 1$, the interpolation parameters contains only the parameters of task $i$, and $\lambda = 0$ contains the parameters of the task $j$.

- We annex the task-specific parameters $\phi_{interp}$ to the global parameters $\theta$ to obtain the complete model $\mathcal{M}$.

- The interpolated samples $\mathbf{x}$ can be generated as:

$$\mathbf{x} = G(\mathbf{z}; \theta, \phi_{interp}) \tag{3}$$

  where $z \in \mathbb{R}^{256}$ is a fixed noise and $z \in \mathcal{N}(0, I)$ and for the unconditional generation $c = 1$. $G(\mathbf{z}; \theta, \phi_{interp})$ is the generator network with the interpolated parameters.

In Figures 4, and 5, we demonstrate smooth interpolations of images between two diverse tasks. Figure 4 shows interpolations of the celebA dataset to a completely diverse flower dataset. In Figure 6, we interpolate between the gray and color scale. The results clearly depict that we can modulate the images of source dataset to the target dataset by making changes in the value of $\lambda_{interp}$, which shows the effectiveness of our approach in transferring the feature map space of one task to another to incorporate continual learning.

We further demonstrate the interpolation results in Figures 7, 8, 9. The results clearly depict that the transfer of target images from source images is smoother and fast, while considering task based similarity for the initialization of target adapters. We can perceptually observe that Anime dataset has more similarity to the CelebA dataset than X-ray dataset, which is evident through the smooth interpolation between CelebA and Anime images.

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

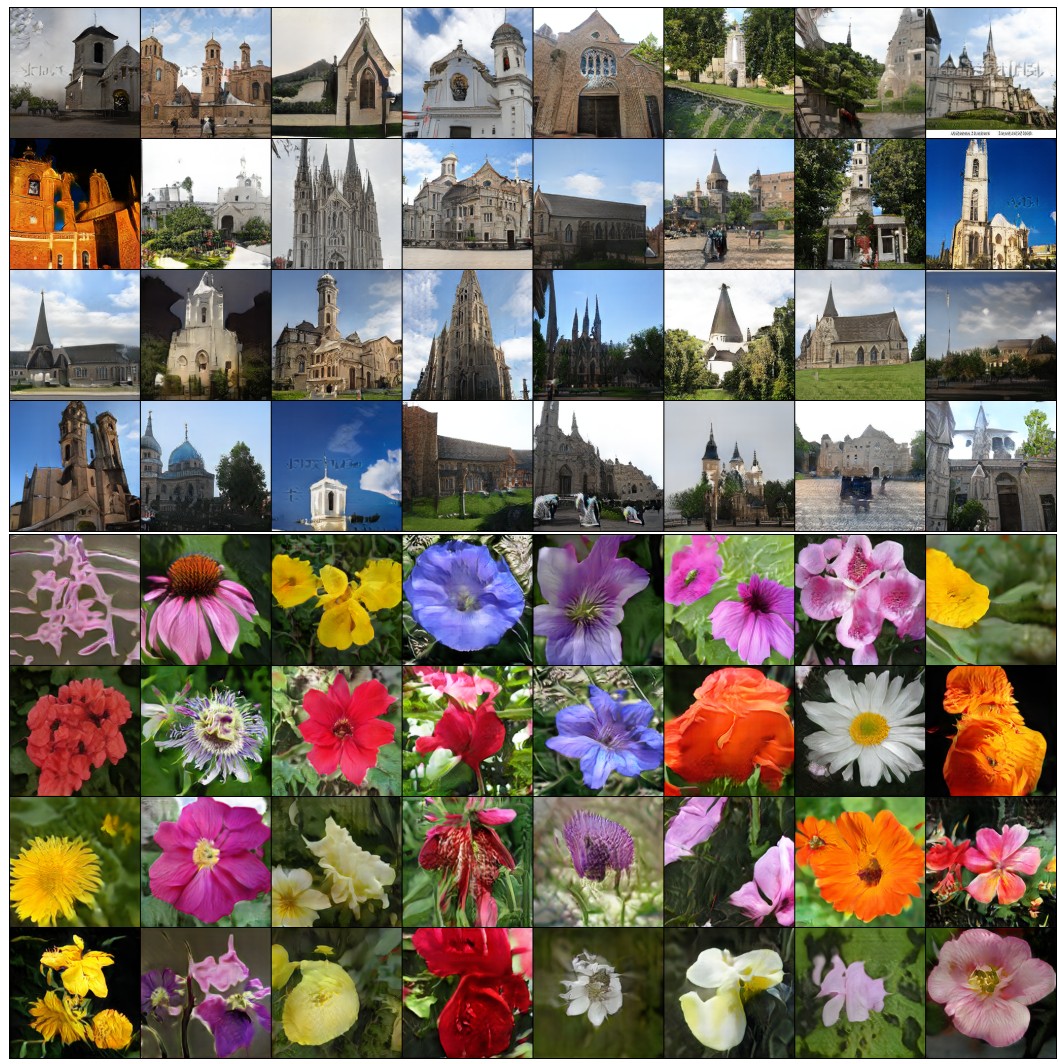

Figure 1: Images generated using our approach, Top:Church Bottom: Flower

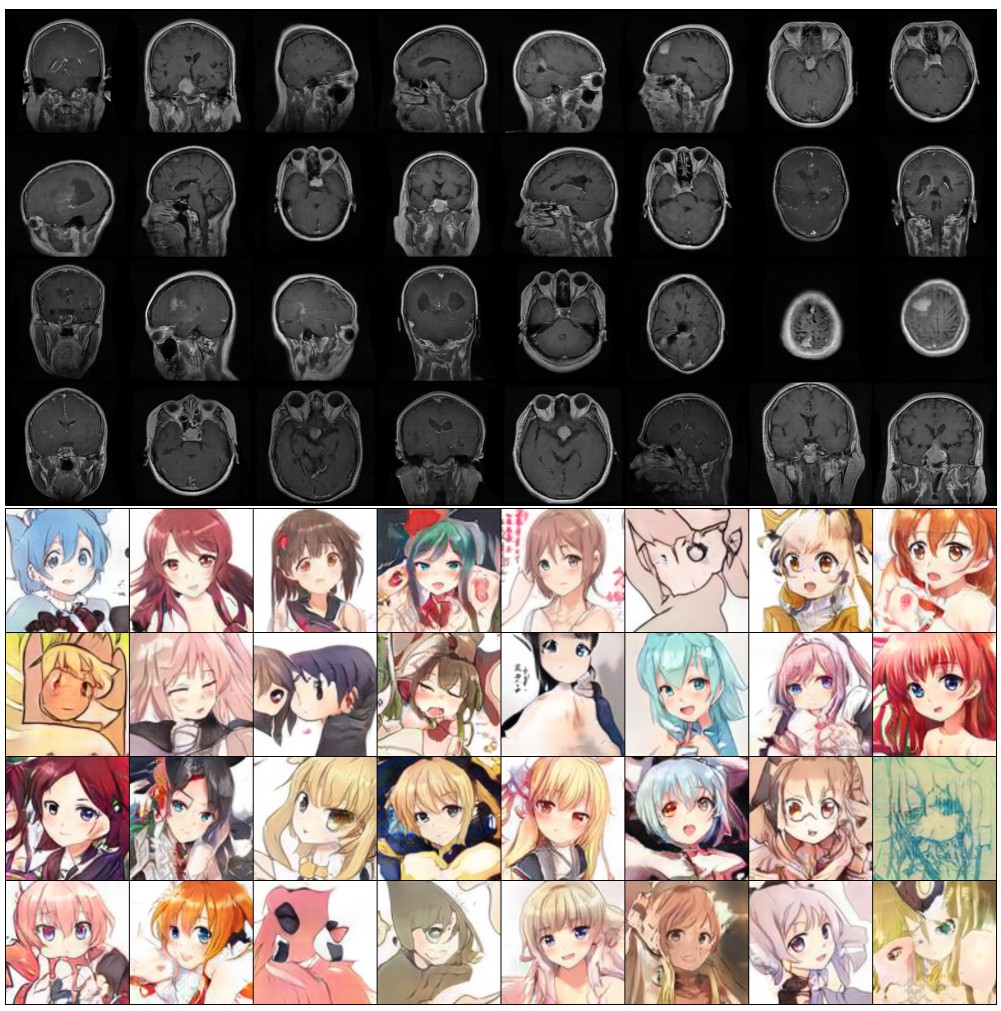

Figure 2: Images generated using our approach, Top: Brain MRI Bottom: Anime faces

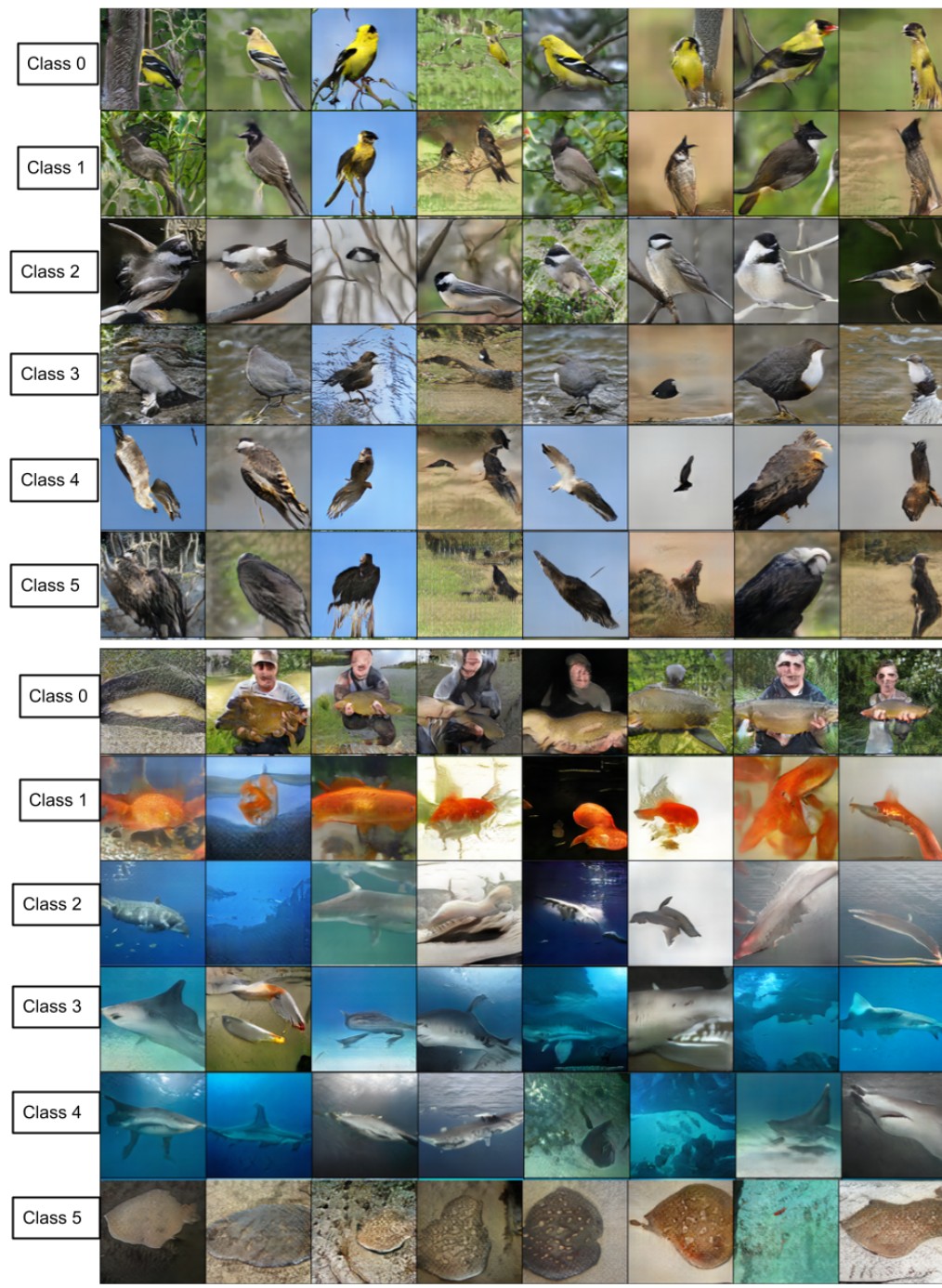

Figure 3: Conditional images generated using our approach for generative replay a)Bird(Top), Fish(Bottom)

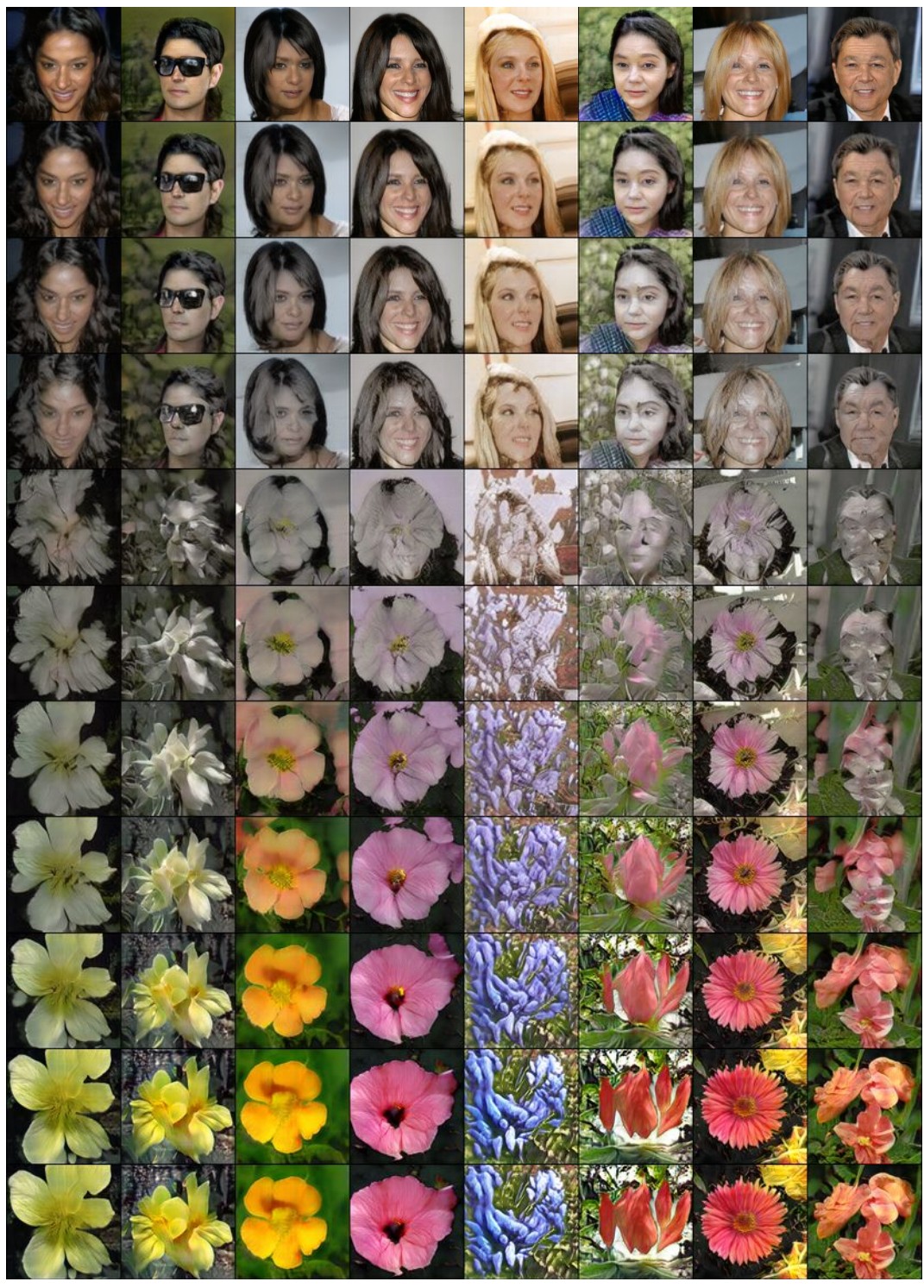

Figure 4: Smooth interpolations of task's parameters between CelebA and Flower images generation using our approach for $\lambda = [0.0, 0.1, 0.2, 0.3, 0.4, 0.5, 0.6, 0.7, 0.8, 0.9, 1.0]$

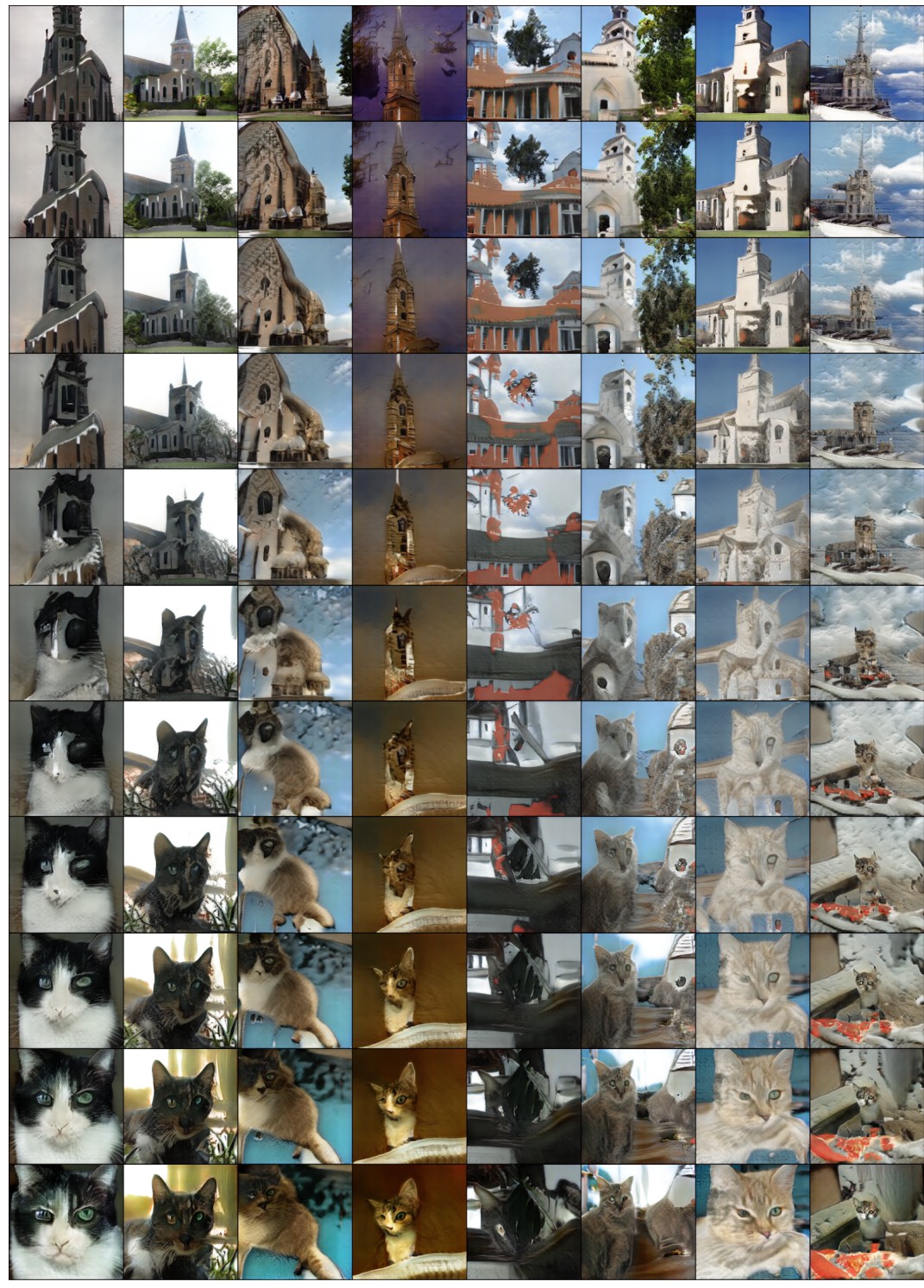

Figure 5: Smooth interpolations of task's parameters between Church and Cat images generation using our approach for $\lambda = [0.0, 0.1, 0.2, 0.3, 0.4, 0.5, 0.6, 0.7, 0.8, 0.9, 1.0]$

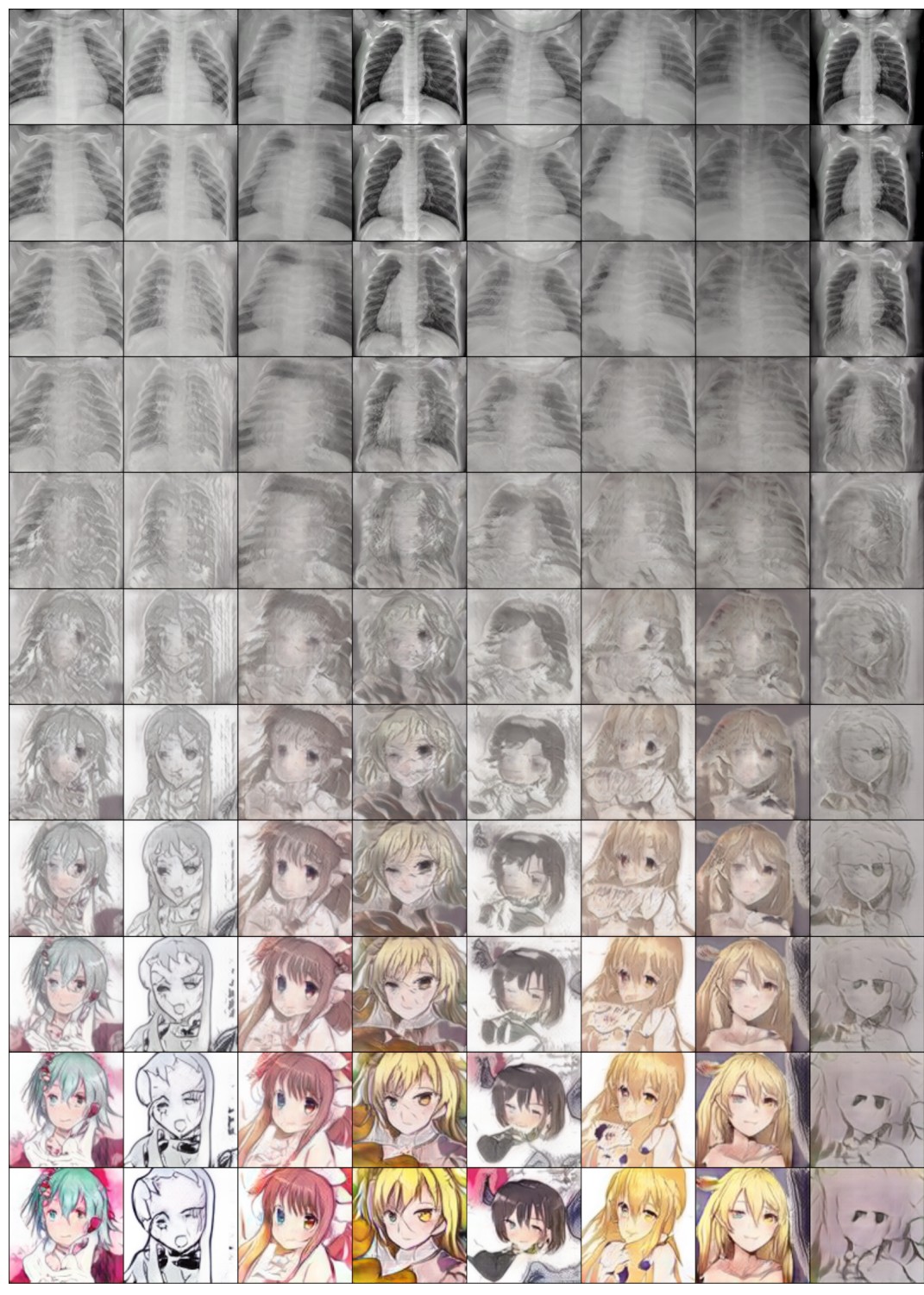

Figure 6: Smooth interpolations of task's parameters between X-ray and Anime images generation using our approach for $\lambda = [0.0, 0.1, 0.2, 0.3, 0.4, 0.5, 0.6, 0.7, 0.8, 0.9, 1.0]$

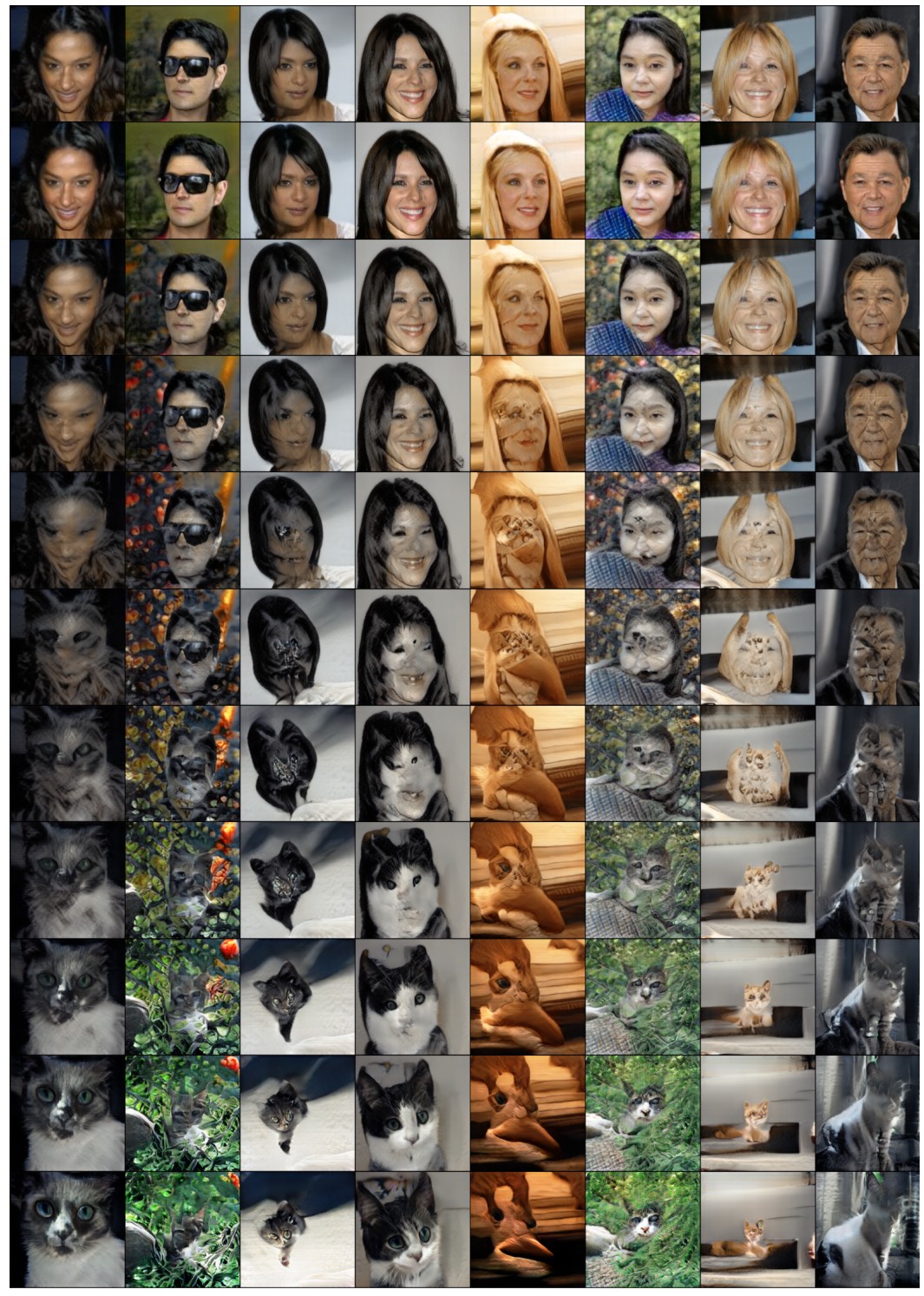

Figure 7: Smooth interpolations of task's parameters between CelebA and Cat images generation using our approach for $\lambda = [0.0, 0.1, 0.2, 0.3, 0.4, 0.5, 0.6, 0.7, 0.8, 0.9, 1.0]$

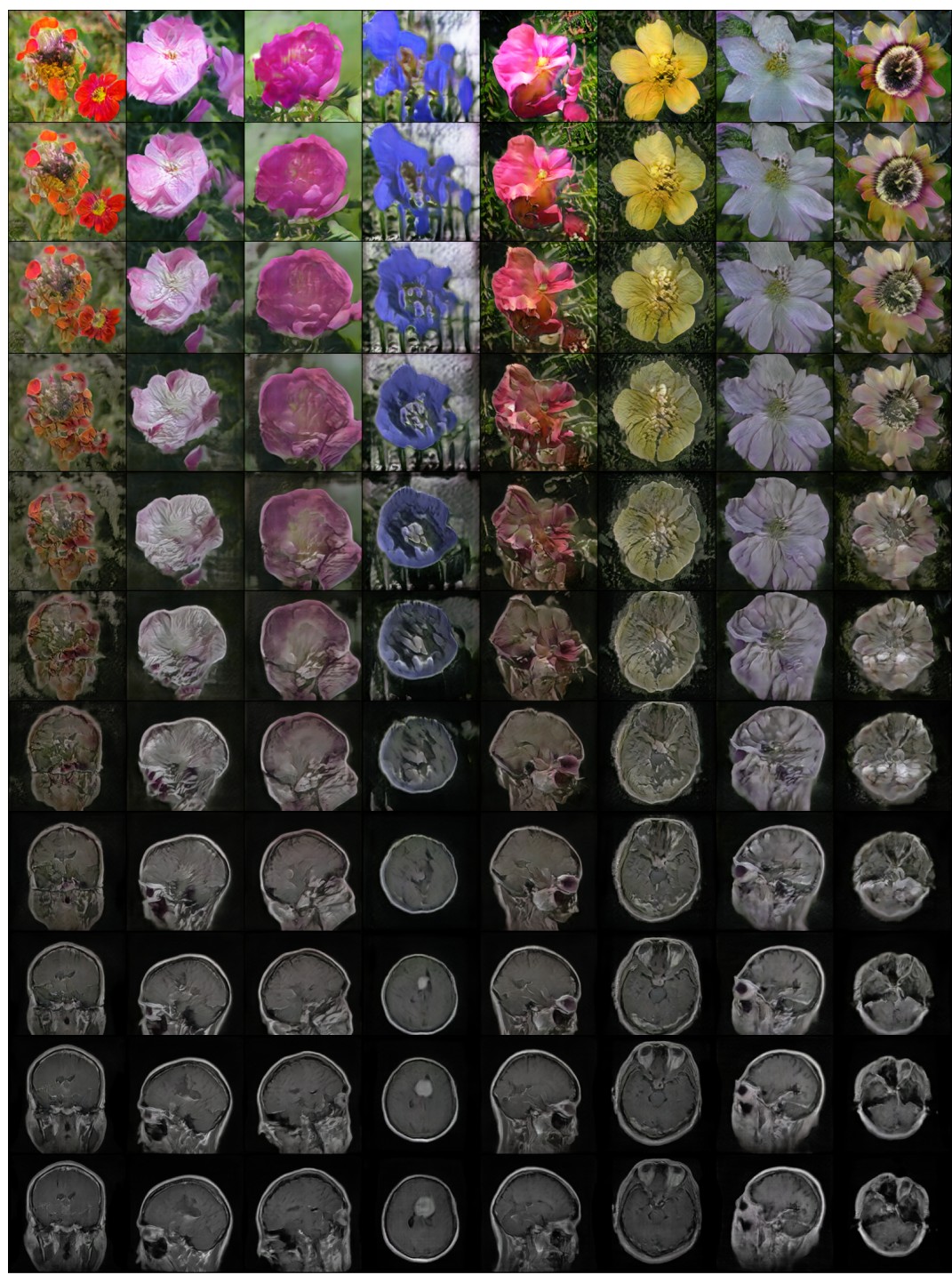

Figure 8: Smooth interpolations of task's parameters between Flower and Brain images generation using our approach for $\lambda = [0.0, 0.1, 0.2, 0.3, 0.4, 0.5, 0.6, 0.7, 0.8, 0.9, 1.0]$

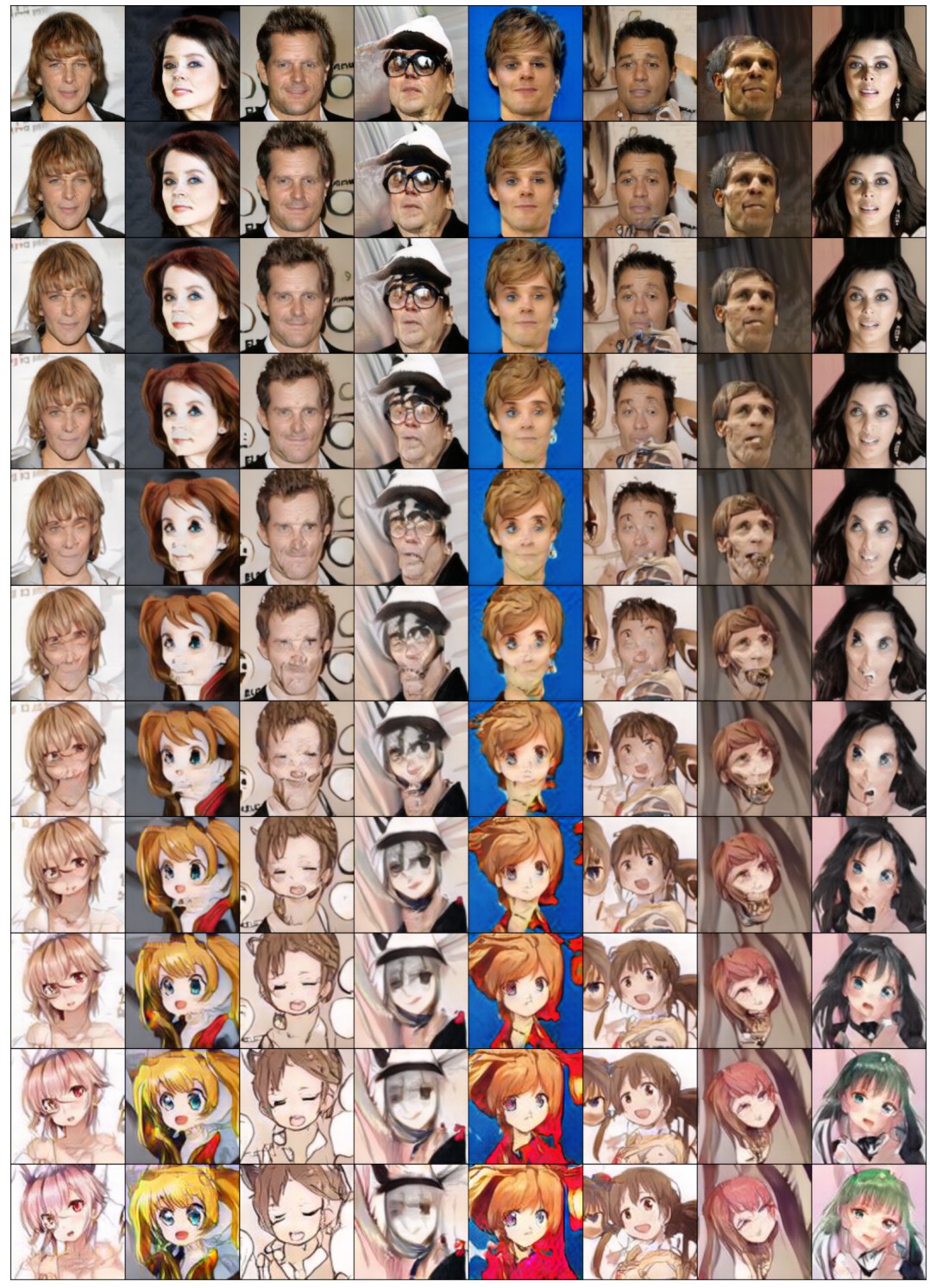

Figure 9: Smooth interpolations of task's parameters between CelebA and Anime images generation using our approach for $\lambda = [0.0, 0.1, 0.2, 0.3, 0.4, 0.5, 0.6, 0.7, 0.8, 0.9, 1.0]$