# OpenReview forum: "CAM-GAN: Continual Adaptation Modules for Generative Adversarial Networks"
_NeurIPS.cc/2021/Conference — NeurIPS 2021 Poster_

### Official Review · Reviewer_1bkf · 2021-07-05

**Rating:** 7
**Confidence:** 5

**Summary:**

The paper proposed to learn a set of global and task-specific parameters to get task-specific feature maps for different tasks. And the proposed method leverages the Fisher information matrix to measure task similarities for parameter initialization for expansion-based continual learning. The proposed method is demonstrated on unconditional generation tasks and label-conditioned generation tasks.

**Limitations And Societal Impact:**

Please refer to the comments in Cons in the main review sections.

**Main Review:**

- Pros
    - The paper is well written and easy to read.
    - The tasks are constructed from different datasets, instead of different categories from same dataset, which is more challenging.
    - The idea of considering task similarities has some merits in it.
    - The proposed approach works well for unconditional generation tasks.


- Cons
    - The contribution is marginal. The idea of considering task similarities share some similarities with [R1], but with key difference. However, the idea of global/task-specific parameters is very similar to [22, 15, R2]. [22] uses task-specific parameters on top of the feature maps to obtain new feature maps for different tasks, while [15] uses the frozen parameters from the initial task then reconstruct task-specific filters from them. The word "global" is bit inappropriate, since same as [15] the "global parameters" are parameters trained on the initial tasks and are frozen for later tasks, and unlike [12, R2] in which parameters are adapted for all later tasks using knowledge distillation. Can authors please explain in details how the proposed approach is **different and better** than [22, 15, R2]?\
    \
    [R1] Expert Gate: Lifelong Learning with a Network of Experts, cvpr'17 \
    [R2] hyper-lifelong gan: scalable lifelong learning for image conditioned generation, cvpr'21\
    $~$
    - Can author please report the average FID over all tasks after the final task is learned for both unconditional and conditional generations?\
    $~$
    - Can proposed approach be applied to image-to-image translation tasks? Would the gradients be similar for tasks like edges2shoes and edges2handbags? What do authors imagine/think the TSL perform will be on these types of tasks?\
    $~$
    - For TSL, what is the performance of random initialization? Given a new task, corresponding task-specific parameters are additional parameters which do not exist in previous models. For a baseline approach, maybe it is more natural to initialize them randomly instead from the most recent task.

**Time Spent Reviewing:**

2-3

---

> ### Author Response · Authors · 2021-08-10
> **Response to Reviewer 1bkf**
>
> We sincerely thank the reviewer for the positive comments. Thanks for the references [R1,R2]. Below, we respond to your questions related to R1 and differences with [22,15,R2]. We hope you will consider raising your scope and supporting our paper (in addition to our response below, we also request you to please look at the comments from reviewers KNMU and s8zU regarding novelty and contribution). If there are any other follow-up questions, we will be happy to answer.
>
> ++ Discussion about [R1, R2, 15, 22] ++
>
> Although the idea of task similarity is also explored in [R1], we want to bring a few key differences to reviewer's notice below. We also discuss the differences with [22], [15], and hyper-lifelong GAN.
>
> [R1] explores the task relatedness in the supervised learning setting where labels for each task are provided, while our approach explores the same in the unsupervised learning setting. Learning the task embedding for unlabelled data is much harder compared to the supervised setting.
> $2):$ Also, for the task embedding, our approach leverages the Fisher Information Matrix (FIM) which primarily focuses on the gradient information while R1 learns the task embedding using the last FC layer. The same will not be possible for the unlabelled data. Reviewer-KNMU and Reviewer-s8zU point to the same novelty.
>
> Progressive Network (PN) [22] is inefficient for model expansion. The PN model has a different parallel network for each task (multi-task), and while training the novel task, they transfer the knowledge from previous tasks by learning lateral connections. For the lateral connection, they compute the linear combination of layer activations from previous tasks and add to the inputs of the corresponding layer for each novel task. Therefore, it has transfer learning from the previous task, but each task learns its own parameters and lateral connection because of the unshared weight. The PN model has polynomial parameter growth [V3 Figure-5 b and Section 4.3 Model Size Comparison]. Therefore PN architecture is not suitable for a large architecture, while our approach is highly scalable for large architecture.
>
> Piggyback-GAN [15] is also an expansion based approach, and it also focuses on efficient model expansion. However, our approach is significantly different compared to Piggyback-GAN. In our approach, we leverage the combination of efficient convolutional architecture to transform each layer's feature map to adapt to the novel task. However, [15] leverages the matrix factorization based approach to adapt to the novel task. For a particular layer $l$, let total filters shape is $h\times w\times C_{i}\times C_{o}$. Then, to adapt to the novel task, it requires $C_{o}\times C_{o}$ size filter. Here $h, w, C_{i}$ and $C_{o}$ are height, width, the number for input filter from the previous layer and number of output filters, respectively.
> In most of the cases (except the boundary of the residual block) $C_{i}=C{o}$ and $h=w=3$; therefore, the parameter on the adapter filter is $9\times$ lower than the base filter ($3\times 3$ replaced by $1\times 1$). Therefore adapter filter contains $\sim11$% parametes as compared to the base model. At the same time, our approach requires $\sim10$% parameters as compared to the base model. Also, [15] leverage over the matrix factorization based approach and for combining the matrix paper adds significant overhead to the model. While our approach can learn large task sequences without significant overhead and lower parameter growth. Also, we observe that compared to the weight transformation based (GAN-Memory) approach feature map transformation is more robust and provides stable training.
>
> Hyper-lifelong gan [22] extend [15], and it leverages a hyper-network [V4] to overcome the task-specific parameter storage. They learn another GAN to generate the task-specific parameters. To learn each task parameter, [22] learns a separate GAN, which is costly. GAN cannot generate the model parameter robustly if the output dimension is too large; also, if the number of task-specific parameters is small, these parameters will not be sufficient to adapt to the novel task. Therefore, to make a balance between the two is a difficult task. Again, knowledge distillation is a costly operation that doubles the model FLOPs during training since the same input has to be passed to the $t^{th}$ and $(t-1)^{th}$ generator. However, our approach does not require $T$ separate GANs for the $T$ tasks. Also, the performance of our approach does not depend on the generation quality of the task-specific parameter. Unlike [22], we do not use knowledge distillation; therefore, we do not require a costly training procedure. We will add a discussion in the final version.
>
> [R1] Expert Gate: Lifelong Learning with a Network of Experts, cvpr'17
>
> [15] Piggyback GAN: Efficient lifelong learning for
> image conditioned generation, ECCV'20
>
> [22] Progressive Neural Networks, Arxiv'16
>
> [R2] hyper-lifelong gan: scalable lifelong learning for image conditioned generation, cvpr'21
>
> [V3] An Adaptive Random Path Selection Approach
> for Incremental Learning, NeurIPS'19
>
> [V4] Continual Learning with Hypernetworks. ICLR'21
>
> =========================
> "Can author please report the average FID over all tasks after the final task is learned for both unconditional and conditional generations?"
>
> For the unconditional:
>
>                  Flower|Cathedral|Cat |Brain-MRI|X-Ray|Anime|Mean-FID
>
> GAN-Memory: 14.8 |   15.6  |  18.2  |  15.6   |17.2  |  13.3  |  15.8
>
> Our (celeb):23.0 | 9.5  |  18.2  |  15.6     |  14.2  |  13.1  |  15.6
>
> Our (Bedroom)24.1| 8.6     |  17.3  |  14.9   |  11.8  |  14.2  |  15.1
>
> The FID and mean across all task for the GAN-Memory and our approach. Our (celeb) and Our (Bedroom) means model use celeb and bedroom as a base task respectively. We provide this result in supplementary please refer to supplementary (Table-2) for more details.
>
> In the limited time for the conditional generation we are able to collect the FID score of Bird and Fish, we are reporting the same and remaining we will provide in the updated version:
>
>             Bird |  Fish   | Mean
>
>  FID   39.7 |  60.1   | 49.9
>
> =============================
>
> "Can proposed approach be applied to image-to-image translation tasks? Would the gradients be similar for tasks like edges2shoes and edges2handbags? What do authors imagine/think the TSL perform will be on these types of tasks?"
>
> We have not tried our approach for the image-to-image translation task, but we believe our proposed expansion based model can be used for the continual image-to-image translation task. Unlike the GAN in the image-to-image translation task; the generator takes one image as input from the image pair. The paired image is easily manageable by changing the input to the generator model. We don't think any constraint that can restrict the model on the paired image.
>
> Without an experiment, it is hard to comment how similar the results will be as compared to edges2shoes and edges2handbags. Because of the time limit and limited resources, we are unable to conduct the experiment during rebuttal period. Also, note that gradient similarity is relative in our case, and it does not provide any significant information if there are just two tasks. If we have $K$ number of tasks, we can measure which task is more similar to which task, and that information can be used for the parameter initialization.
>
> We believe that TSL will be extremely helpful not for continual learning only but the different problems like domain adaption, image-to-image translation etc. The same observation is pointed by Reviewer-2 (s8zU) who stated that: "Using a task-similarity based on Fisher Information scores to select the closest task is an interesting idea that can be adapted by other works in continual learning, domain adaptation, and transfer learning."
>
> ==================================
>
> "For TSL, what is the performance of random initialization? Given a new task, corresponding task-specific parameters are additional parameters which do not exist in previous models. For a baseline approach, maybe it is more natural to initialize them randomly instead from the most recent task."
>
> We have evaluated the random initialization result for the two task sequences Flower and Cathedral; the results are as follows:
>
>        Flower  | Cathedral
>
> FID        35.4   |   20.1
>
> We can clearly see when the adapters are initialized using previous task adapters, based on task similarity or using the previous task sequence, it provides significantly better results as compared to the random initialization.

---

> > ### Comment · Reviewer_1bkf · 2021-08-18
> > **Re: Rebuttal Response**
> >
> > Thanks very much for the detailed discussions regarding the related works [R2, 15, 22]. Some details are incorrect, but I agree with the most advantages of this approach discussed here. I agree and as I mentioned, the idea of TSL is with key difference from [R1], which I appreciate. Thanks for the metrics, and thanks for the ablation study. My concerns regarding these have been addressed.
> >
> > I am interested in TSL for image conditioned generation, as I wonder whether the observation similar tasks share a similar gradient will hold. For unconditional or class conditional generations, input random noises are sampled from the same distribution across all tasks, while for image conditioned generation some tasks share the same input domain while others don't. Similarities are measured (probably?) more regarding the relations between input and output domains. Maybe there could be an extension of this paper exploring TSL for image conditioned generation, which has rarely been explored before.
> >
> > Overall speaking, I would like to raise my score and lean towards accepting the paper.

---

> > > ### Author Response · Authors · 2021-08-18
> > > **Re: Re: Rebuttal Response**
> > >
> > >  Thanks for reading and responding to our response. We are glad that our response addressed your concerns.
> > >
> > > The reviewer said above that "I would like to raise my score and lean towards accepting the paper". However, currently we are unable to see the updated score and we request the reviewer please update the score.
> > >
> > > **About the reviewer's question on image conditioned generation**
> > >
> > > We have not yet tried the similarity measure for image conditioned generation but we believe (though this is mostly based on intuition) that similar tasks will have similar gradient information.
> > >
> > > Intuitively, measuring the similarity for the unconditional generation might be more challenging since the input is sampled from the same distribution and thus the input for each task is indiscriminative. However, for conditional generation, the input will be more discriminative (because of different conditioning distributions). Therefore, for conditional generation, the relationship between the input and output domains may be more discriminative and should be reflected in the gradients. Thus, we believe that for the conditional generation, similar tasks will share similar gradients.
> > >
> > > That being said, it is mostly based on our intuition and it may not be entirely true, but we strongly believe that the same TSL approach should work for image conditioned generation as well. As the reviewer said, it will be a very interesting line of future work to extend our approach for image conditioned generation.

---

> > > > ### Comment · Reviewer_1bkf · 2021-08-19
> > > > **Score Updated**
> > > >
> > > > Thanks for the explanation! The score has been updated.

---

### Official Review · Reviewer_G1J5 · 2021-07-16

**Rating:** 6
**Confidence:** 3

**Summary:**

This paper proposes a lightweight adapter architecture, CAM-GAN, for generative adversarial networks (GAN) and its training method. This adapter enables continual adaptation to new tasks through feature transformation. The proposed CAM-GAN shows the state-of-the-art performance in continual adaptation for various tasks while minimizing the increase in adapter parameters and computation.

**Limitations And Societal Impact:**

This work has no potential negative societal impact.

**Main Review:**

The strength of the proposed method is that it greatly reduces the size and computation of the adapter for continual adaptation. However, there are no big differences from the existing methods regarding adaptation through expansion or adding a separate module for each task. And using 1x1 convolution or group convolution to reduce adapter parameters and computation amount is commonly used in efficient CNN. So its contribution is minimal.
Furthermore, compared with GAN Memory, GAN Memory also includes a parameter compression method, but the comparison is missing.

The paper has several critical typographical errors. In residual learning, element-wise addition of the output of the $(l-1)^{th}$ layer is a shortcut connection, not a residual connection. So, the explanation in Session 2.2 is a bit confusing. And session 5.2, an experiment related to Residual Bias, is not cleared. The "w/o RB" needs to clarify whether it is not using an additional shortcut connection or is using a shortcut connection without $f^r_{\phi^t_{lr}}$. And in order to check the effectiveness of residual bias, it is necessary to compare the case where only shortcut connection is used.

[minor] line 134: $f_g$ > $f^g$

**Time Spent Reviewing:**

7

---

> ### Author Response · Authors · 2021-08-10
> **Response to Reviewer G1J5**
>
> We sincerely thank the reviewer for the helpful comments. Below, we respond to your comments related to the concern about novelty. We hope you will consider raising your scope and supporting our paper (in addition to our response below, we also request you to please look at the comments from reviewers KNMU and s8zU regarding novelty and contribution).  If there are any other follow-up questions, we will be happy to answer.
>
> Below, we respond to each of your questions/concerns:
>
>
> (1) We respectfully disagree about the comment about novelty. Indeed, expansion and adaptation based approaches have been proposed recently. However, efficient model expansion with minimal parameter expansion still remains a key challenge. This aspect has been appreciated by Reviewer-KNMU and Reviewer-s8zU. The proposed feature map transformation-based adaptation is novel, and we have also shown that, using this proposed model, we can have a better or very competitive results than GAN-Memory while our approach uses $50$% fewer parameters. If the reviewer thinks that we have a very close similarity with some specific paper, we request a reference so we can explain the difference with the proposed model as well as include it in the final version.
>
>  Also, as pointed out by Reviewer-KNMU and Reviewer-s8zU, task-similarity (TSL) based initialization is novel, and they appreciate our contributions. In the experiment and ablation, we have shown that TSL is a crucial component of the proposed model. Again, if the reviewer finds that it is not a novel contribution, please share the relevant reference so we can explain the difference with the proposed model as well as include it in the final version.
>
>
> (2) We believe that model compression is complementary, and any CNN model can be compressed using the recent model compression techniques. SVD based model compression is not the contribution of GAN-Memory either; the same approach can also be used to compress our model.
>
> (3) Thanks for pointing out the typographical error; we are happy to correct the error. Also, we will provide a more explicit description of Section-2.2.
>
> In Section-5.3, "w/o RB" means if we do not use shortcut connection in our model. In the shortcut connection we have parameter $f^r_{\phi^t_{lr}}$ it contains $\sim0.2$% parameters for the base network. If we use only shortcut connection and ignore the local parameter, we have only $\sim0.2$% parameter to adapt the novel task, and this parameter is extremely low, and the model will not learn anything. In the experiment, we observe that shortcut connection is essential to train the model smoothly while it does not improve the model performance. If we train the model without a shortcut connection, the model diverges quickly after a few epochs, and the model will not be able to learn anything. The same observation is evident by the training loss of the Generator and Discriminator in Figure-5 (Right), as we can observe that without shortcut connection, we cannot train the model. For more details, please refer to Figure-5 (Right), where we can observe that after 125K iteration generator and discriminator loss diverges without any shortcut connection, while shortcut connection provides very stable training.
>
>
> (4) Thanks for pointing the typo; we will fix it in the final version.

---

> > ### Comment · Reviewer_G1J5 · 2021-08-20
> > **Response to authors**
> >
> > Thanks for the authors' detailed responses.
> > I agree that the compression method used in GAN-Memory is hardly their contribution. However, this does not mean that this paper does not have to compare it to the compressed version of GAN-Memory. It is even more so because the performance of the proposed method is not significantly different from that of GAN-Memory. Instead, there is a contribution to using much fewer parameters. In particular, since the point-wise or group convolution is commonly used for model efficiency, the weight reduction itself does not seem to contribute significantly. If this paper solves the special problems when applying such lightweight techniques to continual adaptation, there will be a more significant contribution.  Nevertheless, without objecting to the opinions of other reviewers, I update the rating because this work is above the acceptance threshold.

---

### Official Review · Reviewer_s8zU · 2021-07-16

**Rating:** 7
**Confidence:** 4

**Summary:**

The paper presents an approach for continual learning in GANs by learning new task-specific modules over a base network. This work explores the contributions of different continual learning approaches for generative models and proposes an expansion-based model which learns new task-specific modules for each new task. These modules augment the features of the base task layer. To prevent instability issues, the model also creates additional residual pathways. To choose the parameters for a new task, a task similarity-based initialization is proposed which initializes the new task network parameters with the learnt parameters of the previous task which is most similar to the current task. Experiments are performed on conditional and unconditional image tasks, and ablations are performed on the effectiveness of the proposed modules.


**Limitations And Societal Impact:**

The authors have touched upon the limitation of the method when a bad base task is chosen to train their base model. However, there are no experiments to demonstrate that (although the paper does have the data). The negative societal impact of this work is not very apparent, although generative models can have harmful societal impacts due to the generation of convincing fake data.

**Main Review:**

[comment]: <> (Originality)

The paper takes inspiration from expansion-based CL models which are dynamically expanding with the number of tasks, and do not suffer from catastrophic forgetting. The closest related work the paper mentions is GAN-memory [1], which also uses task-specific parameters to adapt the base parameters for the given task. Unlike the GAN-memory method which uses a StyleGAN-like framework to modify the weight parameters, the proposed method learns additional lightweight modules (namely, $3\times 3$ and $1 \times 1$ group convolutions) which are applied on top of the base modules to adapt features to the task. The FIM-based task similarity is also seemingly novel.

Here are some of my comments and concerns:

[comment]: <> (Quality)

- One of the potential problems with the proposed method may be w.r.t. domain shift between the base task and the new tasks. Suppose the base task is a classification over medical images. If all the subsequent tasks are for natural images, then the initial filters chosen by the base network will pick out features that are for medical images (in the initial layers) and are very likely non-informative w.r.t natural images. The task-specific layers will try to build upon this set of uninformative features which is expected to have suboptimal performance. GAN-memory does not have this problem since it adapts all base layers by augmenting them (instead of building on top of base features), and the augmentation will be stronger for a completely different task. The paper can investigate the nature of the task-specific features learned from completely orthogonal tasks.
- Moreover, the next base layers are trained on the feature outputs of the previous base layers. Modifying the feature inputs with a task-specific layer will yield a task-specific feature that may have a different feature distribution. For example, $f_{\theta_{l+1}}(.)$ is trained with features $F_l^t$ as inputs (on the base task). Modifying  $F_l^t $ in the previous layer by task-specific networks to obtain $\tilde{F_l^t} = g_\phi(F_l^t)$ as input to the layer $f_{\theta_{l+1}}(.)$ is unintuitive because $f_{\theta_{l+1}}(.)$ expects to extract new features from feature vectors $F_l^t$ and not  $\tilde{F_l^t}$. This may be the reason why the proposed method doesn't work right out of the box and a residual path is added from the previous layer as a "correction term". In Line 176, the paper mentions that the method doesn't work right out of the box because the depth effectively doubles. However, the architecture they adapted from [2] has only a few resnet blocks. Resnets are known to work for large depths [3], therefore the paper's argument is unconvincing. To support this argument (L176), experiments can be run on a smaller network to see if it has a significant stability improvement over the originally chosen architecture.
- What is the effect of choosing a different base task $\mathcal{T}_0$ (like Brain MRI)? How does the performance of GAN-Memory and CAM-GAN change with the choice of the base task? An ablation on this experiment may be very helpful, not just for comparison of the paper, but in general for continual learning approaches.
- In Line 329, "The results are shown ... is significant" - this argument is not a convincing enough argument. However, the FID only decreases from 1-2 points. Bad images are known to have a high FID (>50) and real images are generally in the range of 10-30 [4].
- In Figure 6, results without TSL (50%) can also be added for completeness.

[comment]: <> (Clarity)

- The details of task similarity-based transfer (Section 2.3) is not very well-explained. Specifically, for a given task id $t$, we have a set of learned adapter weights $\phi_1 .... \phi_{t-1}$. How are they used to calculate task similarity with previous tasks. I assume a task representation for all previous tasks needs to be saved as well. Are task embeddings computed for all previous task-weight combinations or only a few? At task $t$, what is the task vector that is compared to all previous representations? These details are missing from the paper. The paper can be more mathematically rigorous in adding these details.
- In Line266, does the paper mean "6-class generation problem"? Since the section is on conditional generation.
- Figure 2 is way too cluttered and the figures are too small. Similarly, Figure 3 is redundant. Since GAN-memory and CAM-GAN are expansion-based approaches with fixed base weights and fixed task weights, the loss plots are going to be horizontal lines. Only MerGAN shows degradation, being a non-expansion-based method. Either more baselines can be added, or just a table-based comparison may also suffice instead of having Figure 3.
- why are FLOPS not mentioned for GAN-memory in Table 1? I believe it should be much lower than the FLOPS for CAM-GAN since only the weights are updated (and conv weights are very small in size compared to the intermediate feature layers), and the effective depth is half of CAM-GAN. This number should also be included for a fair comparison (tradeoff b/w FLOPS and performance).
- In Figure 4 (Right), bar plots can be used instead of line plots.


[comment]: <> (Significance)

The paper demonstrates a new continual learning approach for GAN-based models. It does outperform closely related work, though not by a significant margin. Using a task-similarity based on Fisher Information scores to select the closest task is an interesting idea that can be adapted by other works in continual learning, domain adaptation, and transfer learning.

Ref:

[1] https://arxiv.org/pdf/2006.07543.pdf

[2] https://arxiv.org/pdf/1801.04406.pdf

[3] https://arxiv.org/pdf/2103.07579.pdf

[4] https://arxiv.org/pdf/1706.08500.pdf

**Time Spent Reviewing:**

4

---

> ### Author Response · Authors · 2021-08-10
> **Response to Reviewer s8zU**
>
> We sincerely thank the reviewer for the positive comments.
>
> In our response below, we respond to the various points/questions you had. We have also reported the results related to your question on the choice of base task. We hope you will support the paper and consider raising the score if you deem appropriate. If there are any other follow-up questions, we will be happy to answer.
>
> Our responses below are ordered in the same sequence as the points you asked.
>
> (1) We agree with the reviewer that if the base task is very different (weak) from the subsequent tasks, then adaptation using very few task-specific parameters would be hard. However, we can handle the above scenario up to a certain limit by increasing the amount of task-specific parameters. This would result in increased the model size and FLOPs requirement. Also, we observe that a reasonable base task works well. To support the same claim, we changed the base task from the CELEB dataset to the BEDROOM dataset and trained the model, and we obtained better or very competitive performance. For more details, please refer to Table-2 in supplementary material. Also, for convenience, we provide the same result below:
>
>                   Flower  Cathed.  Cat    Brain-MRI X-Ray    Anime
>
> CAM-GAN (CelebA)        |  23.0    |  9.5     |  18.2    |  15.6    |  14.2    |  13.1
>
> CAM-GAN (Bedroom)     |  24.1     |  8.6     |  17.3    |  14.9    |  11.8     |  14.2
>
>
> As we can observe, that base task is important, but a reasonable base task is sufficient to adapt to novel tasks.
>
> In some settings, one may have access to task embeddings (e.g., task2vec), which can be used to make an informed selection for the first/base task.
>
> To the best of our knowledge, GAN-Memory suffers from the same problem as the proposed model. It also leverages the learned parameters from the first task, and later they do style transfer using the local style parameter for each subsequent task. Please refer to Gan-Memory Eq: $\hat{W}=f_{\Gamma,B}(W)$, $\hat{W}=\Gamma\odot((W-M)/S)+B$, i.e., it transforms the base task weight to the task-specific weight on each layer using the local parameter $\Gamma,B$. However, we have $\hat{F}=f_{\phi}(F)$, i.e., we transform the base task feature map to the task-specific feature map using the local parameter $\phi$. Therefore both might face the same base task initialization problem. The difference is that GAN-Memory transforms the weight while we transform the feature map, and our experimental evaluations show that feature map transformation is more robust and requires significantly fewer parameters than the weight transformation-based model.
>
> (2) We thank the reviewer for the deep insight and suggestion for the possible instability in the model. We agree with the reviewer that the different feature maps obtained by a particular layer may be a potential reason for the instability, and residual paths act as a correction term. We will add the suggested component in the main paper for the possible instability. Also, to verify our claim, we are happy to include the suggested experiment in the main paper; we will update the description based on the obtained result.
>
> (3) The concern raised by the reviewer is that if we have very different base tasks then the model's performance will be largely affected since the learned global parameters contain very different information. We agree with the reviewer that a very different base task will affect the model's performance but in this scenario, the model shows a reasonable result. In this experiment, we consider Brain-MRI as a base task (grayscale medical image) which is very different compared to the natural image (e.g., flower, church, cat etc). We have following result using the Brain-MRI as a base task:
>
>               Flower | Cathedral
> Our (Brain-MRI) 48.02 | 32.06
>
> As we can observe weak base task has a large performance difference but the results are reasonable, in this setting, our approach has 10\% parameter growth (same as a good/reasonable base task). We can improve the model's performance further by increasing the local/task-specific parameters in the model. Because of the time limit, we are unable to conduct the experiment for all datasets, the remaining results on other datasets we will provide in the updated version.
>
> (4) We will remove the statement in line 329 about significant improvement and rephrase it appropriately. Thanks for the suggestion.
>
> (5) We agree with the reviewer. Due to the time constraint and limited resources, we are unable to provide the results without TSL (50\%) in the rebuttal phase. We are conducting the experiment and will provide the same result in the final version.
>
> (6) We apologize if you found Section 2.3 unclear; we provide more details in supplementary material because of the space constraints but also describe some more details below which we will add in the main paper.
>
> Our approach leverages the trained model on the base task to learn the task embedding for each task sequence. Once the model is trained, we pass each sample to the discriminator and calculate the gradient of the discriminator loss. The average gradient across all samples can be used as task embedding but directly using the average makes the embedding size very large (same as the number of parameters in the discriminator). To overcome the above problem, we store the average gradient of each filter; this makes the embedding size the same as number of filters in the model (ignoring the fully connected layer), and in our case, it is nearly 4000 (negligible dimension compared to the total number of gradients). To measure the similarity with the following task, we store the embedding for each task.
>
> To calculate the similarity of the current task from all the previously stored embedding, we need the embedding of the current task. Since we do not have the trained discriminator for the current task, therefore we directly can not follow the above-discussed approach. Therefore, we trained the model for the current task for a few iterations (~2000 iterations); we then consider the partially trained discriminator as a trained discriminator and similarly calculate the task embedding as discussed above. Now we can measure the similarity with all the previous tasks and initialize the model based on the similarity information.
>
> Once the complete training is done, we use a fully trained discriminator to calculate the task embedding and store the task embedding.
>
> (7) In Line-266, its typo the paper means "6-class generation problem", thanks for pointing the same.
>
> (8) Thanks for the suggestion; we will improve the quality of Figure-2. Also, for Figure-3, the corresponding table is provided in supplementary material. Please refer to Table-2 (in supplementary) for more details. We are happy to replace Figure-3 with Table-2 from the supplementary section.
>
> (9) We agree with the reviewer that GAN-Memory requires fewer FLOPs than CAM-GAN. In our approach, we require $\sim35$% more FLOPs while GAN-Memory requires $\sim21$% more FLOPs per task. We will update Table-1 for the FLOPs information.
>
> (10) Thanks for the suggestion. We will update Figure-4.

---

> > ### Comment · Reviewer_s8zU · 2021-08-21
> > **Re: Response to Reviewer s8zU**
> >
> > Hello Authors,
> >
> > Thank you for the detailed response. I went through the comments, and it's good to know that we agree on most aspects of the paper that are strong and that needed improvement.
> >
> > The response addresses my major concerns.

---

### Official Review · Reviewer_KNMU · 2021-07-17

**Rating:** 7
**Confidence:** 4

**Summary:**

This paper deals with the training of GAN in a continual learning setting, i.e. different tasks are sequentially coming. The authors designed learnable task-specific adapters (transformations) for avoid the catastrophic forgetting problem. And task-similarity based initialization is adopted to improve the performance further. Experiment results show that the proposed method achieves better or comparable performance compared to previous state-of-the-arts, but with remarkably fewer parameters.

**Limitations And Societal Impact:**

The authors adequately addressed the potential negative societal impact of their work.

**Main Review:**

Pros:
1) The paper is well-written and easy to follow.
2) The motivation is clear and reasonable. Ablations verify the effect of each proposed module.
3) The proposed method achieves a good trade-off between introducing additional parameters and model performance.

Cons or questions:
1) How does the model performance change with different amount of parameters introduced by the adapters? For example, the effect of choices of channels and groups in the adapters.

2) The global parameters are fixed across tasks. And the authors adopted the parameters learned from the first task as the global parameters. So I just wonder if the performance of continual learning will be largely affected by the characters of the first task, e.g. the similarities between the first task and the following ones. An ablation study and more discussions relating to this may be valuable.

**Time Spent Reviewing:**

2

---

> ### Author Response · Authors · 2021-08-10
> **Response to Reviewer KNMU**
>
> We sincerely thank the reviewer for the positive comments.
>
> You raised two points regarding cons/questions which we respond below. We have also reported the results of the ablation study you suggested. We hope you will support the paper and consider raising the score if you deem appropriate. If there are any other follow-up questions, we will be happy to answer.
>
> (1) "How does the model performance change with different amount of parameters introduced by the adapters? For example, the effect of choices of channels and groups in the adapters."
>
> Response: Indeed; model performance vs amount of adapter parameters would be a vital ablation. Because of the time limit, we are unable to perform this ablation during the rebuttal period since we are required to train from the base model (because of the architectural changes) to all subsequent tasks. The base model itself is required to train 300K-400K iteration. We would add it in the final version.
>
> (2) "The global parameters are fixed across tasks ...... An ablation study and more discussions relating to this may be valuable."
>
> Response: We agree with the reviewer that if the base task is very different (weak) from the subsequent tasks, then adaptation using very few task-specific parameters would be hard. However, we can handle the above scenario up to a certain limit by increasing the amount of task-specific parameters. This would result in increased the model size and FLOPs requirement. Also, we observe that a reasonable base task works well. To support the same claim, we changed the base task from the CELEB dataset to the BEDROOM dataset and trained the model, and we obtained better or very competitive performance. For more details, please refer to Table-2 in supplementary material. Also, for convenience, we provide the same result (FID scores) below:
>
>                    Flower  Cathed.  Cat    Brain-MRI X-Ray    Anime
>
> CAM-GAN (CelebA)        |  23.0       |  9.5        |     18.2       |  15.6       |  14.2       |  13.1
>
> CAM-GAN (Bedroom)     |  24.1       |  8.6        |  17.3        |  14.9        |    11.8       |  14.2
>
> The concern raised by the reviewer is that if we have a very different base task then the model's performance will be largely affected since the learned global parameters contain very different information. We agree with the reviewer that a very different base task will affect the model's performance but in this scenario, the model still shows a reasonable result. In this experiment, we consider Brain-MRI as a base task (grayscale medical image) which is very different compared to the natural image (e.g., flower, church, cat etc). We have following results (FID scores) using the Brain-MRI as a base task:
>
>               Flower | Cathedral
> Our (Brain-MRI) 48.02 | 32.06
>
> As we can observe weak base task does have a large performance difference but the results are still reasonable. In this setting, our approach has a 10\% parameter growth (same as a good/reasonable base task). We can improve the model's performance further by increasing the local/task-specific parameters in the model. Because of the time limit, we are unable to conduct the experiment for all datasets. We will include the remaining results on other datasets in the final version.
>
> As we can observe, the base task is important, but a reasonable base task is sufficient to adapt to novel tasks.
>
> In some settings, one may have access to task embeddings (e.g., task2vec), which can be used to make an informed selection for the first/base task.

---

> > ### Comment · Reviewer_KNMU · 2021-08-20
> > **Re: Response**
> >
> > Thanks for the authors' detailed response. It largely clarifies my concerns. For the question that how the selection of first task affects the performance of continual learning, the authors provided some results when changing the first task. I think it would be better to make a comparison to previous methods to show if CAM-GAN would be more sensitive to the characters of first/base task.

---

### Decision · Program_Chairs · 2021-09-27

**Decision:**

Accept (Poster)

**Comment:**

The paper addresses continual learning for GANs with two major novel techniques: 1) learnable task-specific adapters to combat catastrophic forgetting, 2) a task-similarity measure based on Fisher information matrix accelerated by computational approximations. Although the method appears a bit ad hoc, the experimental results show effective performance, and the rebuttal has addressed the bulk of the concerns (some of which are quite insightful).  I suggest that the authors well incorporate the rebuttal to the paper.  Overall it is a neat and practical approach that is a good addition to the proceedings.